# Widening the landscape of transcriptional regulation of green algal photoprotection

Marius Arend [1,2,3,6], Yizhong Yuan [4,6], M. Águila Ruiz-Sola [4,5], Nooshin Omranian [1,2,3], Zoran Nikoloski [1,2,3] ✉ & Dimitris Petroutsos [4] ✉

Availability of light and $CO_2$, substrates of microalgae photosynthesis, is frequently far from optimal. Microalgae activate photoprotection under strong light, to prevent oxidative damage, and the $CO_2$ Concentrating Mechanism (CCM) under low $CO_2$, to raise intracellular $CO_2$ levels. The two processes are interconnected; yet, the underlying transcriptional regulators remain largely unknown. Employing a large transcriptomic data compendium of *Chlamydomonas reinhardtii's* responses to different light and carbon supply, we reconstruct a consensus genome-scale gene regulatory network from complementary inference approaches and use it to elucidate transcriptional regulators of photoprotection. We show that the CCM regulator LCR1 also controls photoprotection, and that QER7, a Squamosa Binding Protein, suppresses photoprotection- and CCM-gene expression under the control of the blue light photoreceptor Phototropin. By demonstrating the existence of regulatory hubs that channel light- and $CO_2$-mediated signals into a common response, our study provides an accessible resource to dissect gene expression regulation in this microalga.

Photosynthetic microalgae convert light into chemical energy in the form of ATP and NADPH to fuel the $CO_2$ fixation in the Calvin–Benson cycle[1]. They have evolved to cope with rapid fluctuations in light[2] and inorganic carbon (Ci)[3] availability in their native habitats. When absorbed light exceeds the $CO_2$ assimilation capacity, the formation of harmful reactive oxygen species can lead to severe cell damage; this is prevented by the activation of photoprotective mechanisms, collectively called non-photochemical quenching (NPQ). NPQ encompasses several processes that are distinguished in terms of their timescales[2], among which the rapidly reversible energy-quenching (qE) is, under most circumstances, the predominant NPQ component[2,4]. The major molecular effector of qE in the green model microalga *Chlamydomonas reinhardtii* (hereafter *Chlamydomonas*) is the LIGHT HARVESTING COMPLEX STRESS RELATED protein LHCSR3, encoded by the *LHCSR3.1* and *LHCSR3.2* genes[5] that slightly differ only in their

promoters; LHCSR1 can also contribute significantly to qE under conditions where LHCSR3 is not expressed[6,7]. PSBS, the key qE effector protein in higher plants[8] is encoded in two highly similar paralogues *PSBS1* and *PSBS2* in *Chlamydomonas*[9]. They are only transiently expressed in *Chlamydomonas* under high light (HL)[9,10] and their gene products accumulate under UV-B irradiation[11]; while their precise contribution in *Chlamydomonas* photoprotective responses is still a matter of ongoing research, current understanding is that PSBS proteins contribute to photoprotection during HL acclimation of *Chlamydomonas* through both NPQ-independent and NPQ-dependent mechanisms[12].

Intracellular levels of $CO_2$ are modulated by the availability of its gaseous and hydrated forms[3] in the culture media and the supply of acetate, which is partly metabolized into $CO_2$[7,13]. Under low $CO_2$, *Chlamydomonas* activates the $CO_2$-concentrating mechanism (CCM)

[1]Bioinformatics Group, Institute of Biochemistry and Biology, University of Potsdam, 14476 Potsdam, Germany. [2]Systems Biology and Mathematical Modeling Group, Max-Planck-Institute of Molecular Plant Physiology, 14476 Potsdam, Germany. [3]Bioinformatics and Mathematical Modeling Department, Center of Plant Systems Biology and Biotechnology, 4000 Plovdiv, Bulgaria. [4]University of Grenoble Alpes, CNRS, CEA, INRAE, IRIG-LPCV, 38000 Grenoble, France. [5]Present address: Instituto de Bioquímica Vegetal y Fotosíntesis, Universidad de Sevilla-CSIC, 41092 Sevilla, Spain. [6]These authors contributed equally: Marius Arend, Yizhong Yuan. ✉e-mail: nikoloski@mpimp-golm.mpg.de; dimitrios.petroutsos@cnrs.fr

to avoid substrate-limitation of photosynthesis by raising the $CO_2$ concentration at the site of RuBisCO, where $CO_2$ is assimilated[3]. The CCM mainly comprises of carbonic anhydrases (CAHs) and of inorganic carbon transporters. Almost all CCM-related genes are under the control of the nucleus-localized zinc-finger type nuclear factor CIA5 (aka CCM1)[14–16], including the Myb Transcription Factor LOW-CO2-STRESS RESPONSE 1 (LCR1) that controls the expression of genes coding for the periplasmic CAH1, the plasma membrane-localized bicarbonate transporter LOW CO2-INDUCED 1 (LCI1), and the low-CO2 responsive LCI6, whose role remains to be elucidated[17]. CIA5 is also a major qE regulator activating transcription of genes encoding LHCSR3 and PSBS, while repressing accumulation of LHCSR1 protein[7].

LHCSR3 expression relies on blue light perception by the photoreceptor phototropin (PHOT)[18], on calcium signaling, mediated by the calcium sensor CAS[19] and on active photosynthetic electron flow[18–20], likely via indirectly impacting $CO_2$ availability[7]. The critical importance of $CO_2$ in LHCSR3 expression is demonstrated by the fact that changes in $CO_2$ concentration can trigger LHCSR3 expression[21–23] even in the absence of light[7]. Accumulation of *LHCSR1* and *PSBS* mRNA is under control of the UV-B photoreceptor UVR8[11] and PHOT[24,25] and is photosynthesis-independent[20,25]. While LHCSR1 is CO2/CIA5 independent at the transcript level[7,25], PSBS is responsive to $CO_2$ abundance and is under partial control of CIA5[7]. A Cullin (CUL4) dependent E3-ligase[24,26,27] has been demonstrated to post-translationally regulate the transcription factor (TF) complex of CONSTANS (CrCO)[26] and NF-Y isomers[27], which bind to DNA to regulate the transcription of *LHCSR1*, *LHCSR3*, and *PSBS*. The putative TF and diurnal timekeeper RHYTHM OF CHLOROPLAST 75 (ROC75) was shown to repress LHCSR3 under illumination with red light[28].

Here, we employed a large compendium of RNAseq data from *Chlamydomonas* to build a gene regulatory network (GRN) underlying light and carbon responses, and thus reveal the transcriptional regulation of qE at the interface of these responses. The successful usage of RNAseq data to infer GRNs has been demonstrated in many studies[29–31], although the data pose some challenges that require careful consideration. All of the developed approaches to infer a GRN quantify the interdependence between the transcript levels of TF-coding genes and their putative targets; the resulting prediction model serves as a proxy for the regulatory strength that the product of the TF-coding gene exerts on its target(s). It is usually the case that the number of observations (samples) used in building the model is considerably smaller than the number of TFs used as predictors, leading to collinearity of the transcript levels and associated computational instabilities; furthermore, as an artifact of the computational techniques, some of the inferred regulations may be spurious[32–34]. To address these issues, here we took advantage of combining the outcome of multiple regularization techniques and post-processing to increase the robustness of identified interactions[33–35]. In contrast to our approach, the existing predicted GRNs of *Chlamydomonas* either focused on nitrogen starvation[36] or used a broad RNAseq data compendium, not tailored to inferring regulatory interactions underlying responses to particular cues[37]. Moreover, these GRNs were not obtained by combining the outcomes from multiple inference approaches, shown to increase accuracy of predictions[30], and their quality was not gauged against existing knowledge of gene regulatory interactions.

We used an RNAseq data compendium of 158 samples (Supplementary Data 1) from *Chlamydomonas* cultures exposed to different light and carbon supply as input to seven benchmarked GRN inference approaches that employ complementary inference strategies[29,30] to identify activating and inhibiting regulatory interactions. We assessed the performance of each approach based on a set of curated TF-target gene interactions with experimental evidence from *Chlamydomonas*. Based on this assessment, we integrated the outcome of the five best-performing approaches into a unique resource, a consensus network of *Chlamydomonas* light- and carbon-dependent transcriptional

regulation. We used the consensus network to reveal regulators of qE genes and demonstrated the quality of predictions by experimentally validating two of the six tested candidates. We show here that LCR1 regulates not only CCM, as previously reported[17], but also qE by activating the expression of LHCSR3, and demonstrate that qE-REGULATOR 7 (QER7), belonging to the SQUAMOSA-PROMOTER BINDING PROTEIN-LIKE gene family, is a repressor of qE and CCM gene expression. Our work consolidates the extensive co-regulation of CCM and photoprotection[7] based on the untargeted assessment of the obtained genome-scale GRN.

## Results

### Computationally inferred GRN recovered known regulatory interactions underlying qE and CCM in *Chlamydomonas*

We first aimed to employ published and in-house generated RNAseq data sets capturing the transcriptional responses of *Chlamydomonas* to light and acetate availability to infer the underlying GRN. To this end, we obtained data from two publicly available transcriptomics studies of synchronized chemostat wild-type (WT) cultures grown in a 12 h/12 h light-dark scheme and sampled in 30 min to 2 h intervals[38,39]. We combined these with our RNAseq data generated from mixotrophically or autotrophically grown batch cultures of the WT and *phot* mutant acclimated to low light (LL) or exposed to HL (Methods, Supplementary Data 1). These data sets capture the expected expression patterns of the key genes involved in CCM and qE (Fig. 1a, Supplementary Fig. 1) in response to changes in acetate availability and light intensity. Specifically, we found strong upregulation of these genes in the light[7,25], and a marked inhibition of *LHCSR3.1/2* and CCM genes by acetate as previously described[7,40].

We employed these data together with a list of 407 transcription factors from protein homology studies[41,42] (Methods, Supplementary Data 2), as input to seven GRN inference approaches to robustly predict TF-target interactions, as shown in benchmark studies[30]. The employed methods infer both activating and inhibiting TF-target interactions. Benchmarking of GRNs usually relies on ground truth data obtained from ChIPseq or transcriptomic profiling of TF mutants. Since to date, no such comprehensive data set exists for *Chlamydomonas* we manually curated a list of known, experimentally validated regulatory interactions underlying CCM and qE in *Chlamydomonas*[22,26–28], to assess the quality of GRNs inferred by the different approaches. As negative control, we included the reported lack of effect of the SINGLET OXYGEN RESISTANT 1 (SOR1) TF on *PSBS1* transcript levels in diurnal culture[43] as well as a set of four TFs, designated TF1-4, whose knock-out or overexpression did not affect *LHCSR3.1* transcript levels (Supplementary Note 1, Supplementary Fig. 2 and Fig. 1b). When we assessed the predicted ranks, as a measure of confidence assigned to the positive and negative ground truth data, we found that two approaches were clear outliers, showing sensitivity of 0%. More specifically, ARACNE[44] and global silencing[45] are unable to recover any positive literature interactions when using a network density threshold of 10% of all possible TF-TF and TF-target interactions (most prominently observable in the case of LCR1 interactions). Possible reasons for this finding are over-trimming or issues with the validity of the underlying assumptions, as seen in other case studies[46]. Since the presence of most gene regulatory interactions is dependent on environmental stimuli, it is considerably easier to experimentally validate the presence of TF-target interactions than to show that they do not take place[47]. Thus, in inferring a consensus GRN we considered the five approaches that were able to recover positive interactions, namely: Graphical Gaussian Models (GGM), Context Likelihood of Relatedness (CLR), Elastic Net regression, Gene Network Inference with Ensemble of Trees (GENIE3), and Network Deconvolution (Methods).

To obtain insights into the performance of these approaches, we next quantified the variability of ranks for the known TF-target gene

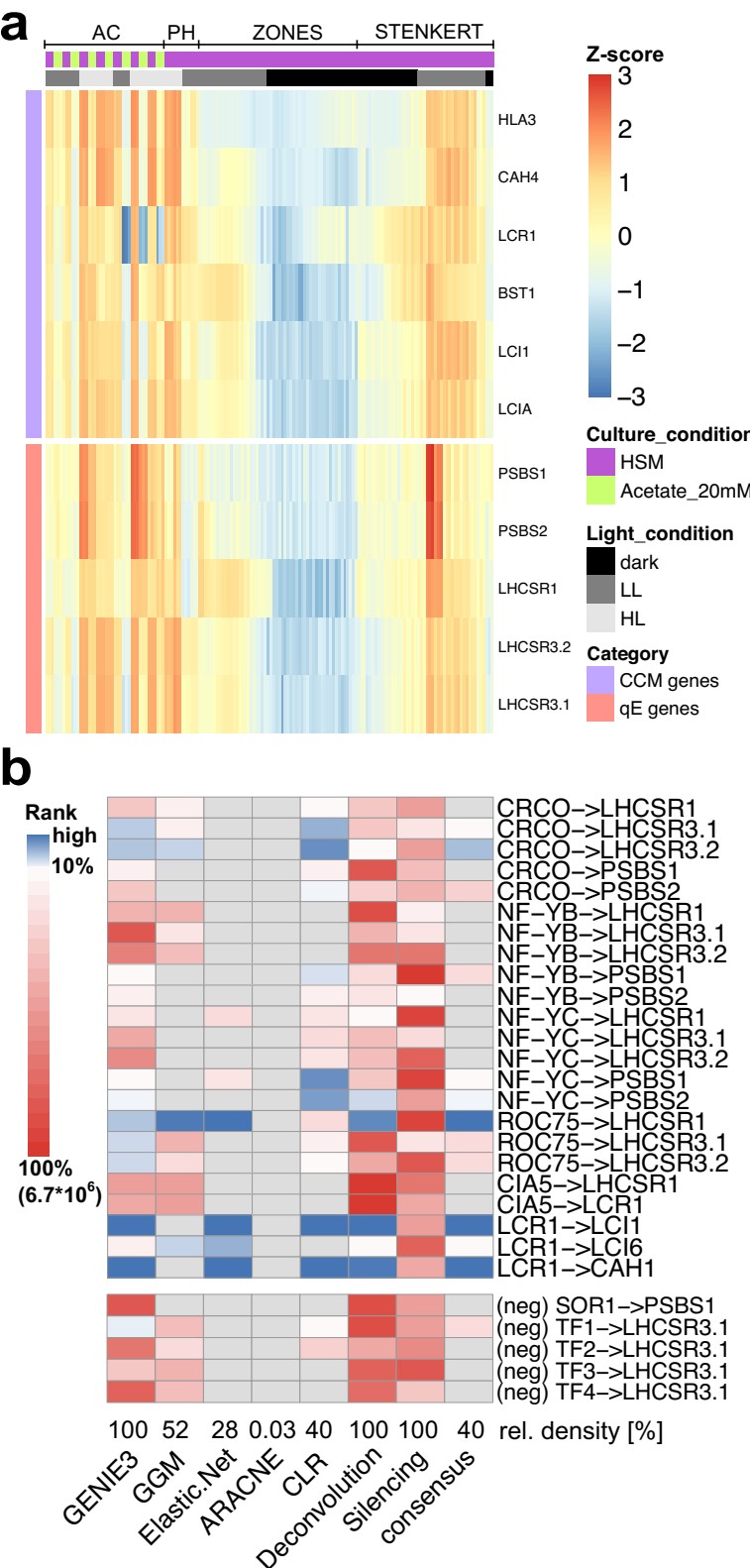

interactions. We found that the average standard deviation of the ranks of the TF-target gene interactions within an approach is larger than the average standard deviation for the rank of a TF-target gene interaction across the five approaches (Supplementary Fig. 3, Fig. 1b). This observation suggested that the properties of a given TF-target gene interaction have a stronger influence on its assigned rank than the inference approach used. More specifically, we noted that the

regulation of LHCSR genes by the two NF-Y paralogues and the induction of LCR1 by CIA5 are not recovered by any of the used approaches; this is in line with reports showing that CIA5 is constitutively expressed and regulated post-translationally[16]—not reflected in the transcriptomics data. Further, NF-Y factors that rely on complex formation with CrCO to regulate their targets[27], may also act via unresolved post-translational mechanisms. As mentioned in the

**Fig. 1 | Characterization of the consensus GRN inferred by employing a compendium of RNAseq data from diverse light and culture conditions.**
**a** Expression levels of representative carbon concentrating mechanism (CCM) and qE genes are plotted over all samples used for network inference (z-scaled log values are depicted, high values red, low values blue); The column annotation gives information on the culture conditions and data set (purple–Sueoka's high salt medium (HSM), green–HSM + 20 mM acetate, light gray–high light (HL), dark gray –low light (LL), black–no light) see also Supplementary Data 1); no clustering was applied. All experimental conditions have been sampled in biological duplicates or triplicates that are plotted adjacent. Supplementary Fig. 1 shows an enlarged version of this figure with sample names included and biological replicates indicated. **b** The heatmap rows correspond to experimentally validated or invalidated (neg)

gene regulatory interactions involved in qE and CCM, curated from the literature. The heatmap indicates ranking of these interactions by different approaches and the consensus network. Edges are considered highly ranked (depicted in blue) if they are above the 10% network density threshold. Edges ranked below this threshold are depicted in red. Edges that were not included in the given network are marked in gray. At the bottom of the heatmap, the relative number of edges inferred by each approach is provided. ARACNE and Silencing columns were only plotted for comparison and were not used in building the consensus GRN (see Methods). TF1 MYB-like DNA-binding protein, Cre01.g034350; TF2 RWP8, RWP-RK transcription factor, Cre04.g218050; TF3 RWP5, RWP-RK transcription factor, Cre06.g285600; TF4 bHLH domain-containing protein, Cre07.g349152). Source data are provided in the Source Data file.

introduction, some of the employed approaches use regularization techniques, mitigating effects of collinearity and low sample number, this comes at the cost of increasing the number of false negatives, another reason for the high number of unrecovered interactions previously reported in the literature. Importantly, the regulatory interactions of the CCM effector genes *LCI1* and *CAH1* by *LCR1* are assigned very high ranks (top 1%) by the approaches considered in the consensus GRN (except for GGM); moreover, only for the interaction of TF1 and LHCSR3.1 we observe false positives, originating from spurious interactions. The interactions of the other four TFs (Supplementary Note 1, Supplementary Fig. 2) are correctly discarded by all approaches, indicating the robustness of the employed approaches (Fig. 1b). In addition, we observed that CLR and GENIE3 demonstrated the best performance with respect to the set of known interactions. For instance, they identified the regulation of *LHCSR3.1* by CrCO[26,27] and of *PSBS1* by NF-YB[27] (Fig. 1b). Generalization of this ranking beyond the known interactions underlying qE and CCM processes is challenging, due to the lack of genome-scale gold standard, and we, therefore, opted to combine the results of the five approaches, that showed comparable performance, in the consensus GRN (Methods, Supplementary Data 3) to increase robustness of the predictions. Our analyses of the overlap between the consensus and individual GRNs and the enrichment of TF-TF interactions demonstrated the robustness of the inferred interactions (Supplementary Note 2, Supplementary Fig. 4).

## Consensus GRN pinpoints LCR1 as a regulator of qE-related genes

Using the consensus GRN, we inferred direct regulators of *LHCSR* and *PSBS* genes and ranked them according to the score resulting from the Borda method (Methods)[30,48]. Mutants were available for four of the top ten of TFs with the strongest cumulative regulatory effect on qE-related genes (Fig. 2a, Supplementary Data 4): Two knock-out mutants of previously uncharacterized genes were ordered from the CliP library[49], which we termed *qE-regulators 4* and *6* (*qer4*, *qer6*; see Supplementary Fig. 5 for the genotyping of these mutants). Additionally, we obtained an overexpressor line of the N-acetyltransferase LCI8[21] and the knock-out strain of the known CCM regulator LCR1[17]. We tested for a regulatory effect by switching LL-acclimated mutant strains and their respective WT background to HL for 1 h and quantified transcript levels of qE-related genes. Since both the paralogs of *PSBS* as well as of *LHCSR3* show correlation >0.96 over all RNAseq samples used in this study and they additionally have very similar expression profiles quantified by RT-qPCR[50], we only probed the transcripts of *LHCSR3.1* and *PSBS1* via qPCR in the validation assays. For *qer4*, *qer6*, and *lci8-oe* we did not observe an effect on the transcript levels of investigated genes after HL exposure (Supplementary Fig. 6). Thus, qer4 and qer6 are considered false positive predictions of the GRN, despite the fact that qer4 accumulated 1.5 times more *LHCSR3.1* under LL than the WT. A review of the closest orthologs of LCI8 together with the experimental data indicate that it is likely involved in arginine synthesis[51] and wrongly included as histone acetylase in the list of TFs.

Interestingly, LCR1, the highest ranking among the tested regulators showed significantly decreased expression of LHCSR3 at both the gene (three times lower, Fig. 2b) and protein level (four times lower, Fig. 2c, d) compared to the WT; as a result, *lcr1* developed very low NPQ and qE (Fig. 2e). Interestingly, the *lcr1* mutant over-accumulated LHCSR1 and PSBS both at the transcript and at the protein level (Fig. 2b–d); Complementation of *lcr1* with the knocked-out gene (strain *lcr1-C*) restored LHCSR3 gene and protein expression as well as the qE phenotype (Fig. 2b–e). Because pre-acclimation conditions impact qE gene expression[25] we conducted independent experiments in which cells were acclimated to darkness before exposure to HL and we obtained very similar results. Our data demonstrated that *lcr1* showed significantly lower expression of LHCSR3 and higher expression of LHCSR1/PSBS at both the gene (Supplementary Fig. 7a) and protein level (Supplementary Fig. 7b, c), and had lower NPQ and qE (Supplementary Fig. 7d) than the WT, although the higher expression levels of the *LHCSR1* gene were not rescued by the complementation with the missing *LCR1* gene (Supplementary Fig. 7a). Altogether, our data show that LCR1 is a regulator of qE by activating *LHCSR3.1* transcription and repressing LHCSR1 and PSBS accumulation.

Furthermore, we revisited the role of LCR1 in regulating CCM genes[17] by analyzing the expression of selected CCM genes in WT, *lcr1* and *lcr1-C* cells shifted from LL or darkness to HL, conditions favoring CCM gene expression[7]. We first confirmed that under our experimental conditions *lcr1* could not fully induce *LCI1* (Supplementary Fig. 8a, b) in accordance with the report of the discovery of LCR1[17]. Our analyses further showed a statistically significant impairment of *lcr1* in inducing genes encoding the Ci transporters LOW-$CO_2$-INDUCIBLE PROTEIN A (LCIA), HIGH-LIGHT ACTIVATED 3 (HLA3), and BESTROPHINE-LIKE PROTEIN 1 (BST1) as well as the carbonic anhydrase CAH4, when shifted from LL or dark to HL (Supplementary Fig. 8a, b), indicating that the role of LCR1 in low-$CO_2$ gene expression extends beyond the regulation of gene expression of *CAH1*, *LCI1*, and *LCI6*[17].

## PHOT-specific GRN reveals a novel repressor of qE

The light-dependent induction of LHCSR3 is predominantly mediated by the blue light photoreceptor PHOT[18]. To analyze the PHOT-dependent transcriptional regulators, we first inferred a GRN based solely on the RNAseq from samples of *phot* and WT acclimated to LL and after 1 h exposure to HL (data set PH, Supplementary Data 1) using only the GENIE3 approach, reported to show good performance[30,46] and which is among the best-performing approaches in our consensus network. Since the PH experiment contains a low number of samples (12 samples, 4 conditions) the inferred GRN will inevitably suffer from the effects of low statistical power and high collinearity. To mitigate these effects, we decided to only include interactions that are also present in the benchmarked consensus network. By determining the intersection of the two networks, we obtained a GRN that resolves regulatory interactions underlying the transcriptomic changes observed in the *phot* mutant, while borrowing the statistical power of the whole RNAseq compendium. We refer to the resulting network as PHOT-specific GRN (Methods, Supplementary Data 5).

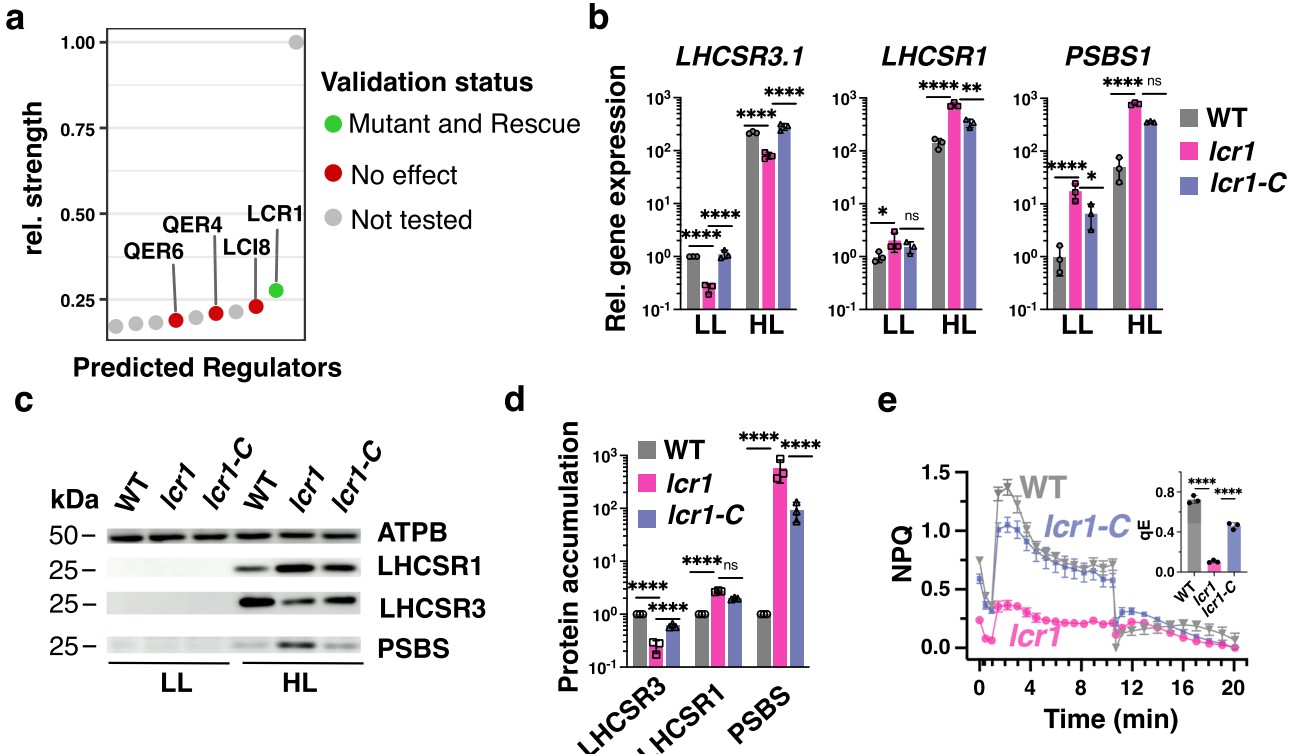

**Fig. 2 | Consensus GRN for light and acetate responses pinpoints LCR1 as regulator of qE-related genes. a** Dot plot of the relative regulatory strength of the top 10 regulators of qE-related genes in the consensus GRN (see Methods). TFs are marked in green if qE transcript levels were affected in the respective knock-out strain and this effect was reversed by complementation with the missing gene. TFs for which no effect was observed are marked in red. TFs for which no mutant lines were available are plotted in gray. **b** WT, *lcr1,* and *lcr1-C* cells were acclimated for 16 h in LL (15 μmol photons m$^{-2}$ s$^{-1}$). After sampling for the LL conditions, light intensity was increased to 300 μmol photons m$^{-2}$ s$^{-1}$ (HL); samples were taken 1 h (RNA) or 4 h (protein and photosynthetic measurements) after exposure to HL. Shown are relative expression levels of qE-related genes at the indicated conditions normalized to WT LL ($n = 3$ biological samples, mean ± sd). **c** Immunoblot analyses of LHCSR1, LHCSR3, PSBS, and ATPB (loading control) of one of the three biological replicates, under the indicated conditions. **d** Quantification of immunoblot data of all replicates in **c** after normalization to ATPB. Shown are the HL-treated samples; WT protein levels were set as 1. **e** NPQ and calculated qE (as an inset) 4 h after exposure to HL ($n = 3$ biological samples, mean ± s.d). **b, d, e.** The two-sided *p* values for the comparisons are based on ANOVA Dunnett's multiple comparisons test and as indicated in the graphs (*$P < 0.005$, **$P < 0.01$, ***$P < 0.001$, ****$P < 0.0001$). Statistical analyses for **b** and **d** were applied on log10- transformed values. The exact p-values are shown in the Source Data file.

When investigating the top 10 regulators of qE genes in this PHOT-specific GRN the interactions where weighted based on the importance score from GENIE3 (Supplementary Data 6, Fig. 3a). Among the interactions in this list of PHOT-dependent regulators, we recovered two known regulators of qE, namely, ROC75[28] and CrCO[26,27] (Fig. 3a). These observations are in line with an existing hypothesis[24] suggesting that a CUL4-dependent E3-ligase targeting CrCO[26] acts downstream of PHOT. ROC75 has been previously reported to act independently of the PHOT signal based on qPCR studies of the mutant grown synchronously under different light spectra[28]. In our RNAseq data, gathered under continuous white light, we observed a significant difference in expression levels of ROC75 between WT and *phot* (log2 fold-change = 1.03, adj. p-value = 1.80*10$^{-7}$).

The fact that several regulators showed larger regulatory strength than CrCO in the PHOT-specific GRN indicates the existence of yet unreported regulators of qE effector genes in the PHOT signaling pathway. This is in line with existing results[26], showing that the knock-out of CrCO is insufficient to fully abolish light-dependent activation of LHCSR3. Following this reasoning we obtained *qer1* and *qer7*, the available regulator candidates mutants, from the CLiP library[49] (for genotyping see Supplementary Fig. 5). Our results show higher mRNA levels of *PSBS1* in the *qer1* mutant (Supplementary Fig. 9); however, this could not be rescued by ectopic expression of the *QER1* gene in the *qer1* mutant background (Supplementary Fig. 9). We found significant upregulation of *LHCSR3.1* gene expression in the *qer7* mutant (1.7

times, Supplementary Fig. 10a) also reflected in higher NPQ (Supplementary Fig. 10b) and qE levels (Supplementary Fig. 10c) which we followed up in more detail. To this end, we ectopically expressed the WT *QER7* gene in the *qer7* mutant and generated the complemented strain *qer7-C* that expressed *QER7* to levels similar to those WT (Supplementary Fig. 5c). As a result, the *qer7-C* strain showed reduced *LHCSR3* gene expression, NPQ, and qE levels as compared with the *qer7* mutant (Supplementary Fig. 10a–c). *LHCSR1* and *PSBS* seemed to be unaffected in the qer7 in these LL to HL transition experiments (Supplementary Fig. 10a). As with LCR1, we also performed dark to HL experiments to further characterize the photoprotective responses of *qer7*; under these conditions, *qer7* accumulated significantly more *LHCSR1* (1.7 times) and *PSBS1* (2.2 times) while *LHCSR3* remained unaffected (Supplementary Fig. 10d). As in the LL to HL experiments (Supplementary Fig. 10b, c) *qer7* showed more NPQ and qE (Supplementary Fig. 10e, f). Complementation of *qer7* with the missing *QER7* gene restored all phenotypes (*LHCSR1, PSBS*, NPQ, qE; Supplementary Fig. 10d–f). These data validated the prediction of QER7 as regulator of qE gene expression (Fig. 3a) and indicated that QER7 regulates different subsets of qE genes depending on the pre-acclimation conditions; *LHCSR3* when preacclimated under LL, *LHCSR1,* and *PSBS* when pre-acclimation occurs in darkness.

Motivated by these findings and given the fact that most of the Chlamydomonas transcriptome undergoes diurnal changes[39] we decided to address the role of QER7 in regulating qE genes under light/

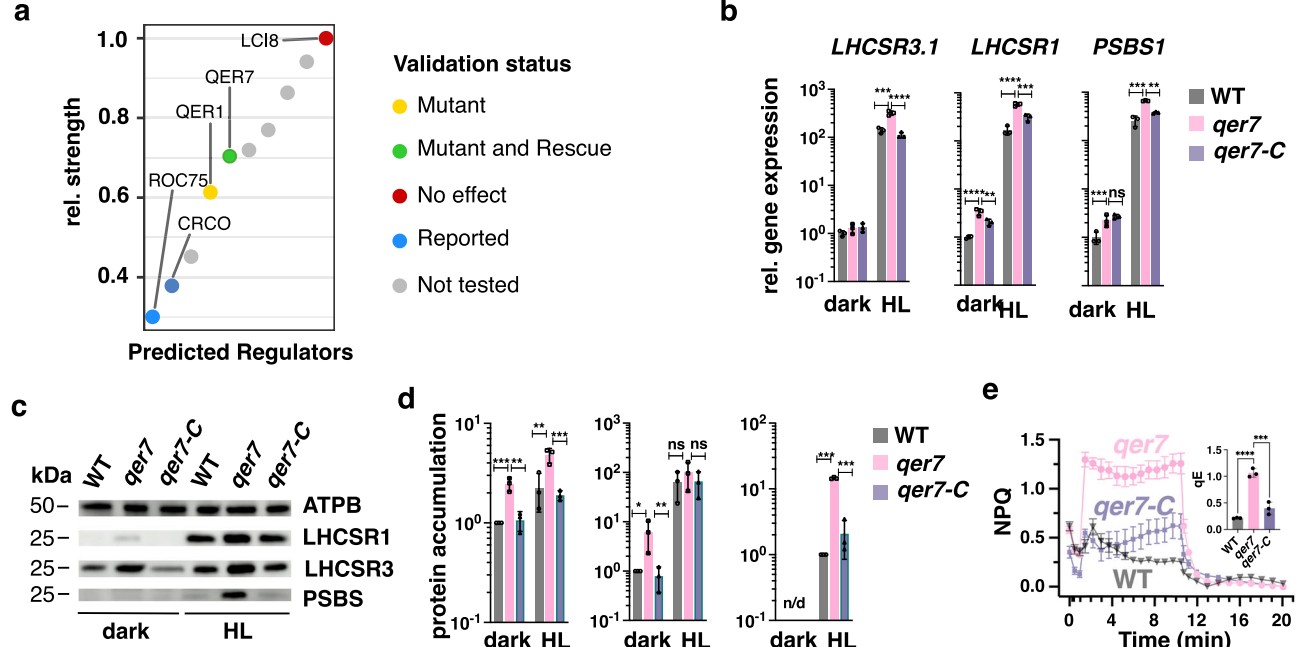

**Fig. 3 | A PHOT-specific GRN pinpoints QER7 as a suppressor of the expression of qE-related genes. a** Dot plot of the relative regulatory strength of the top 10 regulators of qE-related genes in the PHOT-specific GRN (see Methods). TFs are marked in green if qE transcript levels were affected in the respective knock-out strain and this effect was reversed by complementation with the knocked-out gene, in yellow, if the effect was not reversed by complementation and in red, if no mutant effect was observed in the mutant. TFs for which no mutant lines were available are plotted in gray. **b** WT, *qer7*, and *qer7-C* cells were synchronized under 12 h light (15 μmol m⁻² s⁻¹)/12 h dark cycles. After sampling for the dark conditions (end of the dark phase), cells were exposed to 300 μmol photons m⁻² s⁻¹ (HL); samples were taken 1 h (RNA) or 4 h (protein and photosynthetic measurements)

after exposure to HL. Shown are relative expression levels of qE-related genes at the indicated conditions normalized to WT LL ($n = 3$ biological samples, mean ± sd). **c** Immunoblot analyses of LHCSR1, LHCSR3, and ATPB (loading control) of one of the three biological replicate set of samples, under the indicated conditions. **d** Quantification of immunoblot data of all replicates in **c** after normalization to ATPB. WT protein levels at HL were set as 1. **e** NPQ and qE, measured 4 h after exposure to HL ($n = 3$ biological samples, mean ± sd). **b, d, e** The two-sided $p$ values for the comparisons are based on ANOVA Dunnett's multiple comparisons test and are indicated in the graphs (*$P < 0.005$, **$P < 0.01$, ***$P < 0.001$, ****$P < 0.0001$). Statistical analyses for **b** and **d** were applied on log10-transformed values. The exact $p$ values are shown in the Source Data file.

dark cycles. We synchronized WT, *qer7,* and *qer7-C* cells in 12 h L/12 h D cycle and exposed them to HL right after the end of the dark phase. Our results revealed that under these conditions QER7 functions as a repressor of all qE-related genes; the *qer7* mutant expresses significantly higher LHCSR3, LHCSR1, and PSBS not only at the gene (Fig. 3b) but also at the protein (Fig. 3c, d) level, and exhibits higher NPQ and qE (Fig. 3e), with all phenotypes rescued in the *qer7-C* complemented line. Previous protein homology studies identified QER7 as Squamosa Binding Protein[52] or bZIP TF[53], and here, we provide the first functional annotation of QER7 as a novel qE regulator.

### QER7 co-regulates qE-related and CCM genes
Our findings that the regulatory role of CIA5[7] and LCR1 (Fig. 2 and Supplementary Fig. 7) extends beyond CCM to also control qE-related gene expression, prompted us to also inspect the expression levels of CCM genes in synchronized *qer7* cells (Fig. 4a). Indeed, for five of these transcripts (*HLA3, CAH4, BST1, LCI1, LCIA*) we observed a significant upregulation in *qer7* after HL exposure that was reversed by complementation with the *QER7* gene, indicating that QER7 suppresses expression of CCM genes; the suppression role of QER7 on CCM genes was only observable under HL, conditions that favor CCM gene expression[7] and not in the dark (Fig. 4a). We subsequently checked if this regulation is also captured in the GRNs. To this end, we used the CCM genes included in Fig. 4a as target genes and predicted the top 10 regulators of these genes using the same method as for the qE regulators (Methods). Interestingly, we found LCR1 (Supplementary Fig. 11a, c) among the top regulators of both, CCM and qE genes, in the consensus network and QER7 in the PHOT-specific network (Supplementary Fig. 11b, d).

Led by this observation, we investigated the signaling pathway upstream of QER7. To this end, we quantified *QER7* gene expression in synchronized *phot* cultures and observed that *QER7* is overexpressed in the *phot* mutant, suggesting that PHOT suppresses *QER7* expression (Fig. 4b). In contrast to *QER7*, *LCR1* expression levels were WT-like in the *phot* mutant (Fig. 4b), and the same was true for the *qer7* mutant that also expressed *LCR1* to WT levels (Fig. 4a). Further validation that LCR1 and QER7 act on different pathways comes from the fact that although LCR1 is controlled by CIA5[17], QER7 is not (Fig. 4c). Thus, while sharing part of their target genes, the two TFs, LCR1, and QER7, mediate different signals. We captured this distinction in our PHOT-specific network, further underlining the power of the inferred networks.

Our findings that QER7 represses CCM gene expression (Fig. 4a) naturally raised the question of whether the *qer7* mutant has altered CCM function, i.e. altered capacity of the cells to accumulate inorganic carbon (Ci). To address this question, we compared WT and *qer7* cells for their affinity for Ci, under conditions where CCM is not fully induced (LL) and after inducing CCM by acclimation to HL, which leads to mRNA accumulation of CCM-related genes (see for example Fig. 4a but also our previous study[7]). Under these conditions no difference could be observed between WT and *qer7* (Supplementary Fig. 12, Supplementary Data 7). Our data suggest that PHOT, by repressing *QER7* (Fig. 4b), is also involved in the regulation of CCM-related gene expression. We investigated this further by quantifying CCM gene expression in WT, *phot* mutant and the complemented line *phot-C*, in the samples collected from the experiment presented in Fig. 4b, i.e. synchronized photoautotrophic cultures shifted to HL right after the end of the dark phase. We first analyzed expression of qE genes that was found as expected[18,24,25] to be under control of PHOT

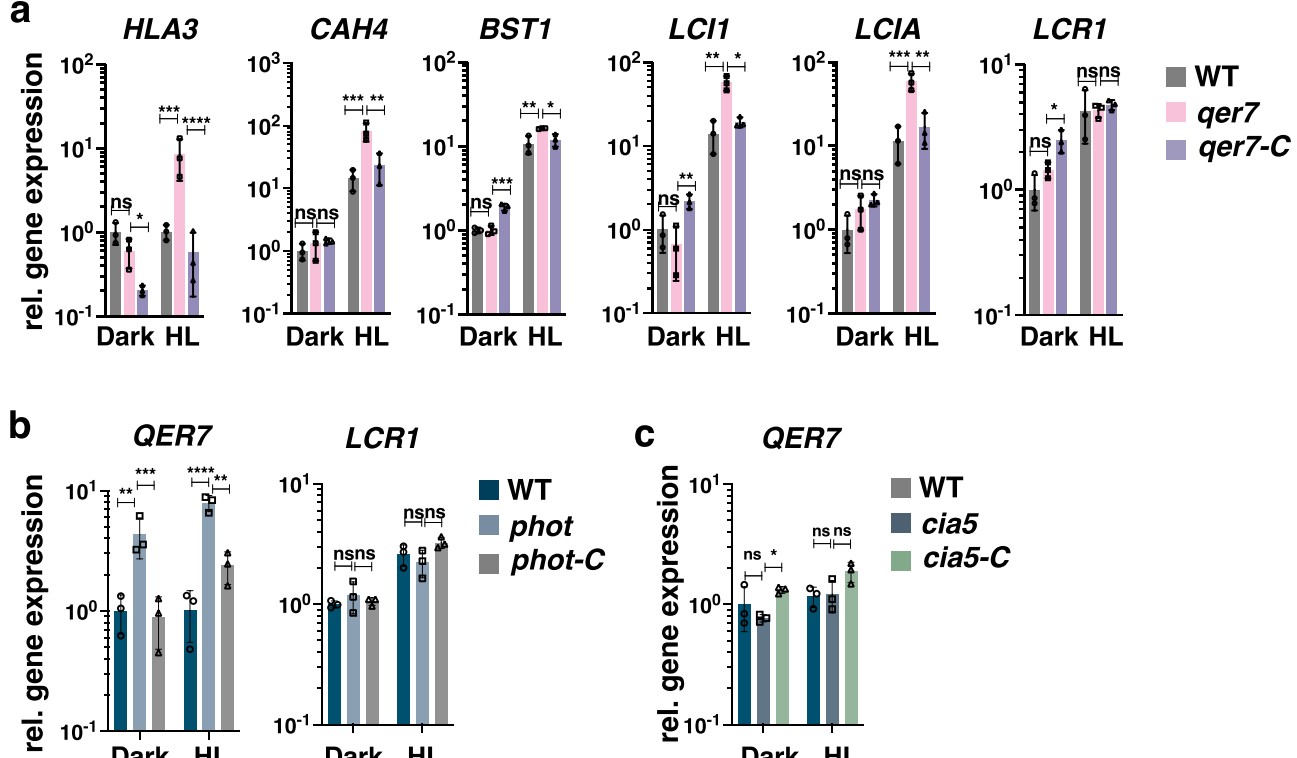

**Fig. 4 | QER7 suppresses transcription of CCM genes and depends on PHOT.** *qer7*, *phot*, and *cia5* cells alongside with their complemented lines (*qer7-C*, *phot-C*, and *cia5-C* respectively) and WT backgrounds, were synchronized under 12 h light (15 μmol m⁻² s⁻¹)/12 h dark cycles, in photoautotrophic conditions (Sueoka's high salt medium; HSM), at 23 °C in Erlenmeyer flasks shaken at 125 rpm. After sampling at the end of the dark phase, cells were exposed to 300 μmol photons m⁻² s⁻¹ (HL) and samples were taken 1 h after HL exposure. **a** Relative expression levels of CCM genes at the indicated conditions. **b** Relative expression levels of *QER7* and *LCR1* in WT, *phot* and *phot-C*. **c** Relative expression levels of *QER7* in WT, *cia5* and *cia5-C*. In all cases (**a**–**c**) expression levels were normalized to WT LL ($n = 3$ biological samples, mean ± sd). The two-sided $p$ values for the comparisons are based on ANOVA employing Dunnett's multiple comparisons test on log10-transformed values and are indicated in the graphs (*$P < 0.005$, **$P < 0.01$, ***$P < 0.001$, ****$P < 0.0001$). The exact $p$ values are shown in the Source Data file.

(Supplementary Fig. 13a). We then analyzed expression of the five CCM genes found to be repressed by QER7 (Fig. 4a); out of those, *CAH4* and *HLA3* were down-regulated in the *phot* mutant exposed to HL after the end of the dark phase in synchronized cultures (Supplementary Fig. 13b) and this phenotype was fully rescued in the *phot-C* complemented line, suggesting a potential involvement of PHOT in regulating expression of CCM-related genes. Nevertheless, the affinity of *phot* for Ci was not different than this of WT (Supplementary Fig. 14, Supplementary Data 7), in line with previous work reporting CCM to be induced to very similar extent under blue or red illumination[54]. Thus, under our experimental conditions, the role of PHOT and QER7 in regulating CCM is restricted to the transcriptional level. Since the PHOT-QER7 pathway acts independently of the LCR1 pathway, both regulating the same subset of CCM genes we tested, it may be not so surprising that neither PHOT nor QER7 impact the affinity for Ci; in both *qer7* and *phot* mutants LCR1 levels are unaffected and therefore the control of LCR1 on CCM may mask any potential effect that PHOT or QER7 might have. Since many known CCM regulatory mechanisms act post-transcriptionally[55], it is conceivable, that the transcriptional regulation of PHOT and QER7 on their own are not sufficient and rely on integration with other simultaneous signals to clear all roadblocks for full CCM induction under HL.

**Genome-scale GRN indicates that photoprotection and CCM are co-regulated**
The two qE regulators that we validated in this study also regulate CCM genes. Therefore, we next investigated to what extent the observed co-regulation pattern applies to the global, known transcriptional regulation of low $CO_2$ and light stress-responsive genes. To this end, we

took advantage of the size of the presented genome-scale GRNs and compiled a list of genes putatively involved in photoprotection (Supplementary Data 8) or the CCM (Supplementary Data 9); we then extracted the 10 TFs exhibiting the strongest regulatory strength on the genes in the compiled lists. We found six (empirical $p$ value < 0.001, Methods) and four (empirical $p$ value <0.01) of the top 10 regulators to be shared between these two responses in the consensus (Fig. 5a, b, Supplementary Data 10) and the PHOT-specific GRN, respectively (Fig. 5c, d, Supplementary Data 11). The significant, large number of shared regulators is a strong indication that co-regulation of photoprotective and carbon assimilatory processes is a principal feature of Chlamydomonas' transcriptional regulatory program.

## Discussion
The molecular actors and structure of the transcriptional regulatory mechanisms that shape *Chlamydomonas'* response to differential light and carbon availability are largely unknown, although they are paramount to survival of *Chlamydomonas* and offer valuable targets for biological engineering. Here we set out to elucidate the GRN underlying the response to light and carbon availability by combining the results from five complementary inference approaches and data from 158 RNAseq samples of cultures responding to these cues. In the network inference process for this study, we carefully choose approaches and integration procedures developed to address the inherent difficulties of RNAseq data sets (e.g., collinearity and high number of variables compared to samples). Together with the many post-transcriptional layers of regulation in eukaryotic cells, the task of recovering the true GRN, nevertheless, remains a major challenge of the field of systems biology and this study is no exception.

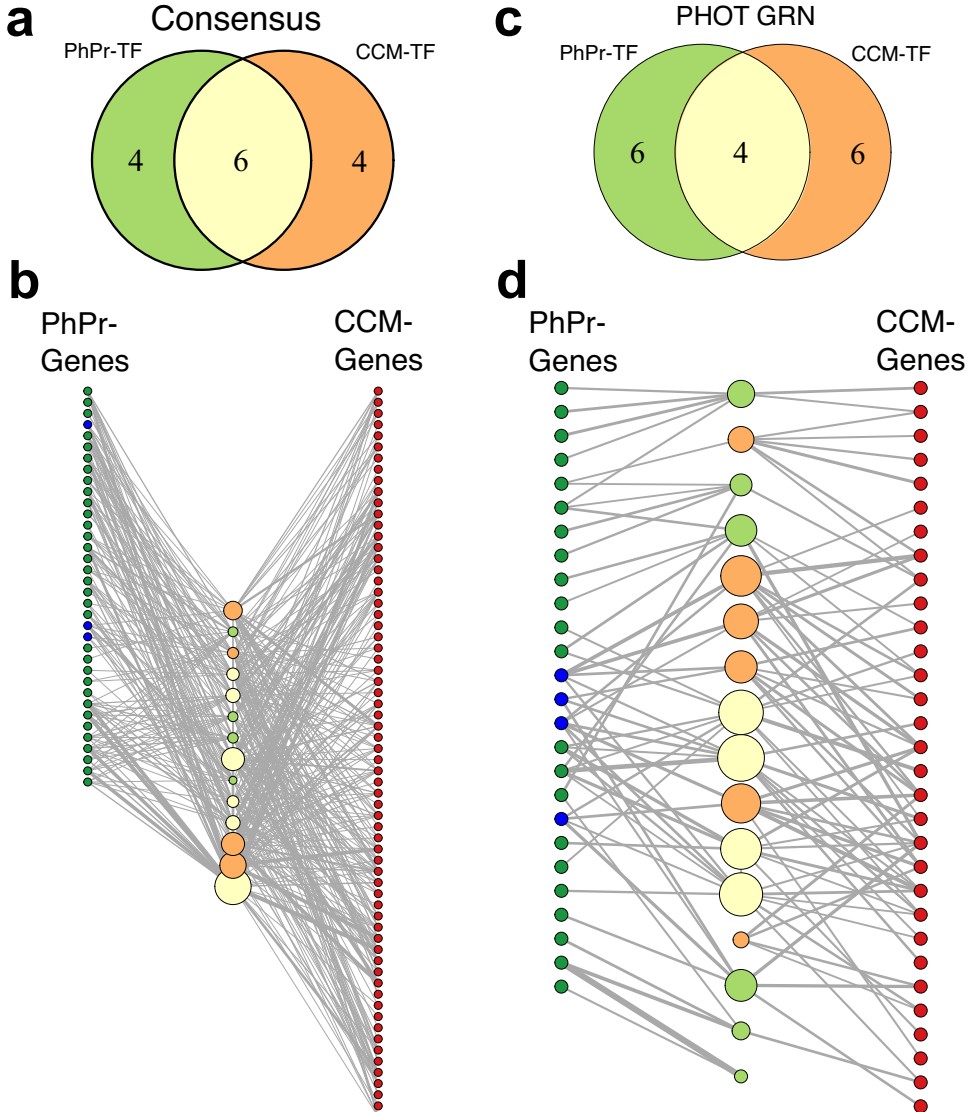

**Fig. 5 | Consensus and PHOT-specific GRNs indicate extensive co-regulation of CCM and photoprotective genes.** Top: Venn diagram depicting the overlap of the top 10 predicted TFs of the curated CCM and photoprotective (PhPr) genes based on **a** the consensus or **c** PHOT-specific GRN. Bottom: Network representation of the top ten TF sets of **b** the consensus network or **d** the PHOT-specific GRN (center nodes, same color code as in **a**) and the target genes used for prediction (left and right columns of nodes, photoprotection genes are shown in green, qE-related genes in blue, and CCM genes in red. **b** The plotted regulatory strength corresponds to $\log(\frac{1}{r_{consen}})$, in **d**, it corresponds the GENIE3 edge weights denoting random forest importance measure. The edge width is proportional to the strength of the specific regulatory interaction. The size of the TF nodes corresponds to the sum of all plotted target gene edge weights.

We were able to experimentally validate two of the six novel qE regulators that the GRN predicted and for which KO mutants were available: QER7, suppressing LHCSR3, LHCSR1, and PSBS expression (Fig. 3, Supplementary Fig. 10), and LCR1, activating LHCSR3 and suppressing LHCSR1 and PSBS expression (Fig. 2, Supplementary Fig. 7). Both TFs also regulate expression of CCM genes, LCR1 as previously reported[17], QER7 as demonstrated in this study (Fig. 4), suggesting that the processes of qE and CCM are co-regulated.

From the physiological point of view, the interconnection of qE and CCM comes as no surprise; exposure to HL boosts $CO_2$ fixation rates and results in depletion of intracellular Ci, reflected, for instance, in the expression levels of the $CO_2$-responsive marker RH1[7]; therefore, HL-exposed cells need to activate not only qE, to protect against photooxidative stress, but also CCM to sustain photosynthetic $CO_2$ fixation. On the other hand, photooxidative damage is exacerbated by $CO_2$-limitation; when $CO_2$ fixation decreases, photosynthetically generated electrons accumulate in the electron transport chain potentially

leading to reactive oxygen species generation[56]. Therefore, acclimation to low-$CO_2$ availability needs to include not only activation of CCM to elevate $CO_2$ levels at the site of fixation, but also of protection against photooxidative damage. It was recently shown that overexpression of the bZIP transcription factor BLZ8, resulting in enhanced CCM via overexpression of HLA3, CAH7, and CAH8, conferred enhanced oxidative tolerance triggered by alkaline stress[57]. Although the qE capacity of the BLZ8 overexpressing lines was not assessed, this work suggests that CCM and protection from oxidative stress are physiologically interconnected.

We complemented the findings of the involvement of LCR1 and QER7 in the regulation of qE and CCM-related genes with an unbiased analysis of the genome-scale co-regulation of CCM and photoprotective genes. In this way we observed a significant number of regulators targeting both processes in the inferred consensus GRN as well as the PHOT-specific GRN (Fig. 5). This finding is in line with several experimental studies: *LHCSR3* mRNA has been reported to

accumulate under low $CO_2$[21,22] while exposure to HL has been reported to trigger CCM protein[58] and mRNA[7] accumulation. The Chloroplast Calcium Sensor protein CAS initially found to be a qE regulator[19] turned out to also regulate CCM[59], and similarly, CIA5, the master regulator of CCM gene expression[14–16] is also controlling qE gene and protein expression[7]. Our results indicate a set of TFs (Fig. 5, Supplementary Data 10, 11) that likely act downstream of these different signals and integrate them into a common transcriptional response. Their further characterization is a promising avenue for future research.

We found QER7 in the list of PHOT-specific regulators and validated its role experimentally: Expression of *QER7* was repressed by PHOT (Fig. 4b) and was CIA5-independent (Fig. 4c), while expression of *LCR1* is regulated by CIA5[17] and was found to be PHOT-independent (Fig. 4b). Thus, we showed that filtering the consensus network by differential expression data of mutants indeed successfully captures this context-specific regulatory interaction. As far as qE is concerned, qE genes are overexpressed in the *qer7* mutant (Fig. 3b), lacking the repressor QER7, and down-regulated in *phot* mutant (Supplementary Fig. 13a), overexpressing the QER7 repressor; as for CCM, QER7 represses expression of all five CCM genes we investigated (Fig. 4a), with two of them, *CAH4* and *HLA3*, being down-regulated in the *phot* mutant under HL (Supplementary Fig. 13b). Further work is needed to obtain a global understanding of the role of phototropin on the transcriptional regulation of Chlamydomonas CCM, including its observed role in controlling some of the CCM genes in the dark (Supplementary Fig. 13b); nevertheless, this suggested link forms an interesting parallel with the convergence of phototropin- and $CO_2$-mediated signals recently shown to control stomata opening, responsible for $CO_2/O_2$ exchanges, in the model plant Arabidopsis[60].

In summary, we presented three valuable sets of PHOT-specific and general regulators: (i) a set of regulators of qE for which we validated available mutants (Figs. 2a and 3a, Supplementary Data 4, 6), (ii) a set of regulators of the core CCM genes analyzed via qPCR in Fig. 4, in which we recovered QER7 and LCR1 as coregulators of expression for qE and CCM genes (Supplementary Fig. 11), and (iii) a set of regulators (Fig. 5, Supplementary Data 10, 11) of genes putatively involved in photoprotection or CCM (Supplementary Data 8, 9), which depicted significant co-regulation at a global scale.

Led by these predictions we experimentally showed that QER7 acts as a repressor of qE and CCM gene expression, LCR1 is a regulator of qE with a more expanded role on regulating CCM as previously thought and finally introduced a photoreceptor-mediated layer of regulation of CCM gene expression. These results clearly demonstrated that the generated GRN represents a powerful resource for future dissection of the transcriptional regulation of responses of *Chlamydomonas* to light and carbon availability. To allow easy access to this resource, we published an R-shiny webtool (https://github.com/arendma/GRN_web) to query the networks for arbitrary regulators and target genes. We expect that the webtool will prompt more concerted, community-wide efforts in resolving the interactions between other pathways that integrate different environmental cues in *Chlamydomonas*.

## Methods
### Transcriptome analysis
We assembled a compendium of RNAseq data (Supplementary Data 1) that capture regulation of light-dependent processes by combining in-house produced RNAseq measurements with publicly available data from two studies of densely sampled diurnal cultures of Chlamydomonas[38,39]. For the samples in the acetate time-resolved experiment, adapter sequences were specifically trimmed from raw reads using BBduk[61] (ktrim = r k = 30 mink = 12 minlen = 50). Raw reads of the diurnal transcriptome study from Strenkert et al.[39] were obtained from NCBI GEO database (GSE112394). Reads were aligned to the Chlamydomonas reference transcriptome[62] available from JGI

Phytozome (Assembly version 5) using RNA STAR aligner. The BAM files obtained from these measurements were analyzed using HTSeq-count[63] (stranded=reverse) to create raw read count files. The raw read counts from Zones et al.[38] were obtained as.tsv from NCBI GEO (GSE71469). The final data set consists of 158 samples from 62 experimental conditions or time points (Supplementary Data 1). Genes with less than 1 count per million in at least 9 measurements where discarded and the remainder were voom[62] transformed and normalized using library normalization factors based on the TMM[64] approach as implemented in the R Bioconductor package edgeR[65].

### Transcription factor set from comparative genomics
To reduce the set of parameters in our network model, we compiled transcription factor (TF) annotations for the Chlamydomonas genome based on proteome homology studies. We obtained the proteomes and protein IDs of predicted Chlamydomonas TFs from Pérez-Rodríguez et al.[41] Since these predictions were built based on the older Chlamydomonas assembly, we first used the conversion table provided by Phytozome to convert JGI4 to Crev5.6 IDs. For the TFs that could not be recovered by this approach we used the Phytozome BLAST tool to align these sequences against the Crev5.6 proteome (BLASTP, E threshold: −1, comparison matrix: BLOSUM62, word length: 11, number of reported alignments: 5). The reported hits were filtered for sequence identity >97% and gaps ≤1 . If sequences mapped multiple times to the same Crev5.6 gene ID, only the hit alignment closest to the N-terminus of the query sequence was kept. The hit was only accepted if the alignment started at least six residues from the N-terminus of the hit sequence. For Crev5.6 loci that had multiple JGI4 TF queries assigned to them the best hit was selected manually. This set was then extended by the TFs found in the study of Jin et al.[42] and the regulators in the manually curated set of CCM and qE regulatory interactions (Supplementary Data 8, 9). Using this procedure, we compiled a list of 407 Chlamydomonas TFs (Supplementary Data 2) to be considered as regulators in the inferred networks.

### Gene regulatory network inference
The CLR and ARACNE approach were based on all replicate measurements; for all other inference methods the median from each condition was used as input. All input matrices where standardized gene-wise. If not explicitly stated in the respective paragraph the implementations of all GRN inference approaches were applied with their default settings.

**Graphical Gaussian models.** The network inferred from a Graphical Gaussian model of gene regulation was obtained using the implementation of the partial correlation estimate from Schäfer et al.[34] as implemented in the R GeneNet package. All interactions between TFs and another gene/TF with non-zero partial correlations were included as network edges.

**GENIE3.** The random forest-based network from GENIE3 was generated using the R Bioconductor implementation provided by the authors[66]. We used only expression levels of TFs as predictors.

**Elastic net regression.** A linear regression-based network was obtained using the elastic net algorithm[35]. A model was fit for each gene using the expression levels of all TFs as predictors. The two hyperparameters $\lambda2$ (quadratic penalty) and s (fraction of L1 norm coefficients) were tuned for each gene model using 6-fold cross-validation. The 2D parameter space scanned was $\lambda2 = \{0, 0.001, 0.01, 0.05, 0.1, 0.5, 1, 1.5, 2, 10, 100\}$ and $s = \{0.1, 0.2, 0.3, 0.4, 0.5, 0.6, 0.7, 0.8, 0.9\}$. The $R^2$ value for each model was calculated as

$$R^2 = 1 - \frac{\sum(\mathbf{y} - \hat{\mathbf{y}})}{\mathrm{var}(\mathbf{y})}, \tag{1}$$

with $y$ denoting the vector of observed expression values and $\hat{y}$ the model predictions. Models with a negative coefficient of determination ($R^2$) value were discarded as regularization artifacts. The results of the remaining models were assembled into a network in which interactions were ranked by regression coefficients $\beta$ normalized by the maximum absolute coefficient:

$$\widetilde{\boldsymbol{\beta}} = \frac{1}{\max\limits_{i}\{|\boldsymbol{\beta}_i|\}}\boldsymbol{\beta}. \qquad (2)$$

**CLR and ARACNE.** The implementation of mutual information (MI)-based network inference approaches from the R package minet[67] was used. Pairwise MI was estimated based on the Spearman correlation as proposed by Olsen et al.[68]. Two networks were constructed based on these MI estimates. Using the CLR approach[69] non-significant interactions were removed based on the z-scores calculated from the marginal distributions of MI values for each gene pair. Alternatively, the ARACNE algorithm[44] was used to prune the network based on the data processing inequality. For both networks only interactions originating from a TF were taken into consideration and edges were ranked according to the assigned MI value.

**Deconvolution and silencing.** For the two networks based on decomposition of the interaction matrix $G$ the Pearson correlation matrix obtained from gene expression values were used as input.

The deconvolution approach introduced by Feizi and co-workers[70] was implemented as previously described[46]. The eigenvalue scaling factor $\beta$ was initialized as $\beta = 0.9$ and iteratively reduced in increments of 0.05 until the largest eigenvector of the direct interaction matrix generated by deconvolution was smaller than 1. Edges were ranked according to the deconvoluted interaction matrix.

The Silencing approach as described by Barzel et al.[45] was implemented in R. The proposed approximation of the direct interaction matrix $S$ in which spurious interactions are silenced relies on the inverse of the observed correlation matrix $G$. In our implementation, we used the Moore-Penrose pseudoinverse in case $G$ was close to singular. In the resulting network, edges were ranked according to the approximated silenced interaction matrix.

**Consensus network construction.** To improve network quality[30], we built a consensus network integrating the GRN models inferred by the different approaches introduced above. To this end, we used the Borda count election method[48] whereby the rank $r$ of an interaction $I$ in the consensus network built on the predictions from $k$ approaches is given by the arithmetic mean of the ranks in the individual networks

$$\mathbf{r}_{\text{consens}}(I) = \frac{\sum\limits_{i=1}^{k}\mathbf{r}_i(I)}{k}. \qquad (3)$$

Following the reasoning of Feizi et al.[70] in this integration only the top 10% all possible edges in the GRN (625815) were considered from each individual ranking. For an edge that was not assigned a rank by some approaches, the missing ranks were set to 10% of all possible edges plus one.

Using this integration method, we assembled a consensus network based on all approaches to compare predictions from all GRN inference approaches (Supplementary Fig. 4b). Due to this comparison and their sensitivity of 0% (Fig. 1b) the rankings derived from ARACNE and Silencing were only considered in Supplementary Fig. 4b and excluded from the final consensus network used for all other analyses. As with the individual networks returned by the different approaches the consensus network (Supplemental Data 3) was trimmed to the top 10% of all possible edges according to the integrated ranks. For

predictions of regulators the weight of edges in the final network was set as $r_{\text{consens}}^{-1}$, denoting the reciprocal of the interaction rank.

**PHOT-specific network**
To investigate the PHOT-specific regulatory interactions genes that are differentially expressed between phot mutant and wt under low and high light were inferred. To this end, transcript counts of genes with more than one count per million in at least four replicates from these conditions (Supplementary Data 1) were tested for differential expression using the R packages limma[71], DeSeq2[72], and edgeR[65]. Only genes deemed significant by all three tools after Benjamini-Hochberg correction for a false discovery rate of 0.05 were considered differentially expressed with respect to PHOT mutation.

In the next step, we focused on the normalized and scaled expression levels from these differentially expressed genes and the previously mentioned conditions, to infer a PHOT-specific GRN using GENIE3. To improve robustness of this network, which was obtained from a comparably sparse data set, we only considered the edges in the intersection with the final consensus network. Again, for both networks only the top 10% of possible edges were taken into account. Therefore, the obtained PHOT-network represents a subnetwork of the final consensus in which edges are weighted by the PHOT-specific GENIE3 importance measure (Supplementary Data 5).

**Identification of major regulators**
We compiled a manually curated list of possible target genes known to be involved in the processes of qE (*LHCSR1, LHCSR3.1/2, PSBS1/2*), photoprotection (Supplementary Data 8), and CCM (Supplementary Data 9). Based on the assumption that major regulators act on several genes important for a biological process, the regulatory strength of a candidate regulator (for the given process) was determined by the sum of edge weights $w_{ij}$ between this regulator and the $k$ genes in the respective target gene set

$$C(TF) = \sum_{j=1}^{k} w_{TFj}. \qquad (4)$$

**Empirical p-value calculation using Monte-Carlo simulation**
The one-sided p-value for the overlap between the regulators of CCM and photoprotective genes was approximated by sampling the overlaps of random gene sets. To this end, we compiled two gene sets with the same cardinality as the curated CCM and photoprotective genes. The genes in these sets where randomly sampled without replacement from all targets in the respective networks. The 10 strongest regulators of these two gene sets where then obtained as previously described and the overlap was calculated as our sample statistic. This process was repeated 10,000 times and an empirical p-value was calculated from the number of iterations, $r$, where the overlap was higher or equal to the observed value, and the total number of iterations, $n$[73]:

$$p = \frac{r+1}{n+1}. \qquad (5)$$

**Strains and conditions**
*C. reinhardtii* strains were grown under 15 µmol photons m$^{-2}$ s$^{-1}$) in Tris-acetate-phosphate (TAP) media[74] at 23 °C in Erlenmeyer flasks shaken at 125 rpm. For all experiments cells were transferred to Sueoka's high salt medium (HSM)[75] at 1 million cells mL$^{-1}$ and exposed to light intensities as described in the text and figure legends. For the investigation of the impact of acetate on the genome-wide transcriptome, HSM was supplemented with 20 mM sodium acetate. *C. reinhardtii* strain CC-125 mt+ was used as WT. The *phot* mutant (depleted from

*PHOT1*; gene ID: Cre03.g199000), *was* previously generated[76] and recently characterized together with its complemented line *phot-C*[25]. The *cia5* mutant (defective in *CIA5*, aka CCM1; geneID: Cre02.g096300; Chlamydomonas Resource Center strain CC-2702), was previously generated[14] and was used along with its complemented *cia5-C* (ref. 7). For synchronized cultures, the cells were grown in HSM for at least 5 days under a 12 h light/12 h dark cycle (light intensity was set at 15 μmol photons m$^{-2}$ s$^{-1}$; temperature was 18 °C in the dark and 23 °C in the light). All CLiP mutant strains used in this study and their parental strain (CC-4533) were obtained from the CLiP library (REF); *qer1* (LMJ.RY0402.072278), *qer4* (LMJ.RY0402.202963), *qer6* (LMJ.RY0402.162350), *qer7* (LMJ.RY0402.118995). The *lcr1* (strain C44), *lcr1-C* (strain C44-B7) and its parental strain Q30P3 as described in[17] were a kind gift from Hideya Fukuzawa. Before performing phenotyping experiments, we first confirmed that *lcr1* shows no expression of *LCR1* and that this is rescued in the *lcr1-C* strain (Supplementary Fig. 15a). The *lci8* overexpressing line was purchased from the Chlamydomonas Resource center; strain CSI_FC1G01, expressing pLM005-Cre02.g144800-Venus-3xFLAG in the CC-4533 background. Overexpression of LCI8-FLAG was verified by immunoblotting against FLAG (Supplementary Fig. 15b).

To complement *qer1*, a 1152 bp genomic DNA fragment from *Chlamydomonas* CC-4533 was amplified by PCR using KOD hot start DNA polymerase (Merck) and primers P11 and P12 (Supplementary Data 12). To complement *qer7*, a 5755 bp fragment DNA fragment from *Chlamydomonas* CC-4533 was amplified by PCR with Platinum superfii DNA Polymerase (Thermo Fisher Scientific) and primers P13 and P14 (Supplementary Data 12). The PCR products were gel purified and cloned into pRAM118[77] by Gibson assembly[78] for expression under control of the *PSAD* promoter. Junctions and insertion were sequenced and constructs were linearized by EcoRV before transformation. Eleven ng/kb of linearized plasmid[79] mixed with 400 μL of $1.0 \times 10^7$ cells mL$^{-1}$ were electroporated in a volume of 120 mL in a 2-mm-gap electro cuvette using a NEPA21 square-pulse electroporator (NEPAGENE, Japan). The electroporation parameters were set as follows: Poring Pulse (300 V; 8 ms length; 50 ms interval; one pulse; 40% decay rate; + Polarity), Transfer Pluse (20 V; 50 ms length; 50 ms interval; five pulses; 40% decay rate; +/- Polarity). Transformants were selected onto solid agar plates containing 20 μg/ml hygromycin and screened for fluorescence by using a Tecan fluorescence microplate reader (Tecan Group Ltd., Switzerland). Parameters used were as follows: YFP (excitation 515/12 nm and emission 550/12 nm) and chlorophyll (excitation 440/9 nm and 680/20 nm). Transformants showing high YFP/chlorophyll value were further analyzed by real time qPCR.

Unless otherwise stated, LL conditions corresponded to 15 μmol photons m$^{-2}$ s$^{-1}$ while HL conditions corresponded to 300 μmol photons m$^{-2}$ s$^{-1}$ of white light (Neptune L.E.D., France; see ref. 7 for light spectrum).

## DNA Isolation and genotyping of CLiP mutants

Total genomic DNA from CLiP mutants and corresponding wild-type strain CC-4533 was extracted according to the protocol suggested by CLiP website (https://www.chlamylibrary.org/). One μl of the extracted DNA was used as a template for the PCR assays, using Phire Plant Direct PCR polymerase (Thermo Fisher Scientific). To confirm the CIB1 insertion site in the CLiP mutants, gene-specific primers were used that anneal upstream and downstream of the predicted insertion site of the cassette (primer pairs P3-P4, P7-P8, P9-P10, and P5-P6 for *qer6*, *qer1*, *qer7*, and *qer4* respectively; Supplementary Data 12). While all these primers worked in DNA extracted from WT, they did not work in the DNA extracted from the mutants, with the exception of *qer4* (Supplementary Fig. 5), therefore primers specific for the 5′ and 3′ ends of the CIB1 Cassette were additionally used. All the primers used for genotyping were shown in Supplementary Data 12. We further confirmed

the disruption of the genes of interest by quantifying their mRNA accumulation (Supplementary Fig. 5).

## RNAseq analysis

For RNA sampling, CC-125 cells grown in TAP at 5 μmol photons m$^{-2}$ s$^{-1}$ were transferred to HSM at 1 million cells mL$^{-1}$ and were left to acclimate to the new medium for 24 h always at 5 μmol photons m$^{-2}$ s$^{-1}$. At time point 0, while maintaining the light intensity at 5 μmol photons m$^{-2}$ s$^{-1}$, cells were either supplied with 20 mM of sodium acetate or remained in HSM and were samples at $t = 15$, 240 and 1440 min. Additionally, at time points 240 min and 1440 min part of the cultures (HSM and HSM supplied with acetate) were exposed to 300 μmol photons m$^{-2}$ s$^{-1}$ and were sampled after 15 min and 60 min exposure to high light (AC experiment; Supplementary Data 1). In an independently designed experiment *phot* and CC-125 cells acclimated for 24 h in HSM and 5 μmol photons m$^{-2}$ s$^{-1}$ at a cell density of 1 million cells mL$^{-1}$, were sampled right before and after 60 min of exposure to 300 μmol photons m$^{-2}$ s$^{-1}$ (PHOT experiment; Supplementary Data 1). Freshly collected samples were lysed using Tri-reagent (Sigma). After double chloroform extraction the aqueous phase was transferred to a RNeasy Mini-Kit column (Quiagen) and processed according to the manufacturers guide. Libraries were generated using the TruSeq LT libraries stranded mRNA protocol and sequenced on a Illumina HiSeq 4000 device.

## RT-qPCR analysis

Total RNA was extracted using the RNeasy Mini Kit (Qiagen) and treated with the RNase-Free DNase Set (Qiagen). 1 μg total RNA was reverse transcribed with oligo dT using Sensifast cDNA Synthesis kit (Meridian Bioscience, USA). qPCR reactions were performed and quantified in a Bio-Rad CFX96 system using SsoAdvanced Universal SYBR Green Supermix (Bio-Rad). The primers (0.3 μM) used for qPCR are listed in Supplementary Data 13. A gene encoding G protein subunit-like protein (GBLP)[80] was used as the endogenous control, and relative expression values relative to *GBLP* were calculated from three biological replicates, each of which contained three technical replicates. All primers using for qPCR (Supplementary Data 13) were confirmed as having at least 90% amplification efficiency. In order to conform mRNA accumulation data to the distributional assumptions of ANOVA, i.e. the residuals should be normally distributed and variances should be equal among groups, Two-way analysis of variance were computed with log-transformed data $Y = \log X$ where $X$ is mRNA accumulation[81].

## Immunoblotting

Protein samples of whole cell extracts (0.5 μg chlorophyll or 10 μg protein) were loaded on 4–20% SDS-PAGE gels (Mini-PROTEAN TGX Precast Protein Gels, Bio-Rad) and blotted onto nitrocellulose membranes. Antisera against LHCSR1 (AS14 2819, 1:15000 dilution), LHCSR3 (AS14 2766, 1:15000 dilution), ATPB (AS05 085, 1:15000 dilution) were from Agrisera (Vännäs, Sweden). Antiserum against PSBS was from ShineGene Molecular Biotech (Shanghai, China) targeting the peptides described in Ref. 9 (used at a dilution of 1:1000). ATPB was used as a loading control. An anti-rabbit horseradish peroxidase-conjugated antiserum was used for detection at 1:10000 dilution. Mouse monoclonal antibody against FLAG was purchased from Sigma-Aldrich (F3165, St. Louis, MO, USA) and was used at a dilution of 1:15000. An anti-mouse horseradish peroxidase-conjugated antiserum (Jackson Immuno Research Europe LTD) was used as a secondary antibody for 3xFLAG immunoblotting (1:10000 dilution). The blots were developed with ECL detection reagent, and images of the blots were obtained using a CCD imager (ChemiDoc MP System, Bio-Rad). For the densitometric quantification, data were normalized with ATPB.

## Fluorescence-based measurements

Fluorescence-based photosynthetic parameters were measured with a pulse-modulated amplitude fluorimeter (MAXI-IMAGING-PAM, Heinz-Waltz GmbH, Germany). Prior to the onset of the measurements, cells were acclimated to darkness for 15 min. Chlorophyll fluorescence was recorded during 10 min under 570 $\mu$mol m$^{-2}$ s$^{-1}$ of actinic blue light followed by finishing with 10 min of measurements of fluorescence relaxation in the dark. A saturating pulse (200 msec) of blue light (6000 $\mu$mol photons m$^{-2}$ s$^{-1}$) was applied for determination of Fm (the maximal fluorescence yield in dark-adapted state) or Fm' (maximal fluorescence in any light-adapted state). NPQ was calculated as (Fm− Fm')/Fm' based on[82]; qE was estimated as the fraction of NPQ that is rapidly inducible in the light and reversible in the dark.

## CO$_2$-dependent O$_2$ evolution

The measurements were performed in accordance with[83] with minor modifications. Cells in photoautotrophic conditions (Sueoka's high salt medium; HSM[75]), shaken in Erlenmeyer flasks at 125 rpm and 23 °C were shifted from LL (overnight at 15 $\mu$mol m$^{-2}$ s$^{-1}$) to HL (4 h at 300 $\mu$mol m$^{-2}$ s$^{-1}$) in order to induce the CCM. Cells were suspended in 4 ml 25 mM HEPES-KOH buffer (pH 7.3), at 25 $\mu$g chlorophyll per mL and were briefly sparged with nitrogen gas to remove the dissolved inorganic carbon (Ci). The cells were then transferred to an oxygen respiration vial (Pyro Science GmbH, Aachen, Germany) and were illuminated at 300 $\mu$mol photons m$^{-2}$ s$^{-1}$ for about 10–20 min, until no net oxygen evolution was seen, an indication that the internal Ci was depleted. Ci concentration in the cell suspension was then increased by stepwise injecting NaHCO$_3$ with a microsyringe. O$_2$ evolution was measured using a fiber optic probe (FireStingO2, Pyro Science GmbH, Aachen, Germany). Cumulative concentration of NaHCO$_3$ after each addition were as follows: 25, 50, 100, 250, 500, 1000, 2000 $\mu$M. K$_{1/2}$(Ci), the Ci concentration needed for half maximal rate of oxygen evolution, and Vmax were calculated by non-linear curve fitting to the Michaelis-Menten equation, using Prism Graph (GraphPad Software, LLC).

## Statistics and reproducibility

Public and in-house generate RNAseq data contained three biological replicates for each sampled condition except for the study of Zones et al.[38] hat reported two biological replicates per conditions. All other experimental results presented in this study are based on three biological replicates as indicated in the respective sections. No statistical method was used to predetermine sample size. No data were excluded from the analyses. The experiments were not randomized. The Investigators were not blinded to allocation during experiments and outcome assessment.

## Reporting summary

Further information on research design is available in the Nature Portfolio Reporting Summary linked to this article.

## Data availability

The in-house RNAseq data generated in this study (Supplemental Data 1) have been deposited to the GEO database under accession codes GSE227473 (phot mutant screen) and GSE227281 (HSM and acetate light stress time course). Other previously published RNAseq data used in this study[38,39] are available in the GEO database under accession codes GSE112394 and GSE71469. The consensus and PHOT-specific GRN generated in this study are provided in edge list format in the Supplemenary Information (Supplemental Data 3, Supplemental Data 4). To allow easy access to the information, we developed an R-shiny webtool that allows to query arbitrary TFs and target genes for regulatory interactions. The R-shiny webtool can be accessed at https://github.com/arendma/GRN_web[84]. The source data underlying Figs. 1–5 and Supplementary Figs. 1–3, 5c, 6–10, 12–15 are provided as a Source Data file. The Source Data file also contains uncropped and

unprocessed scans of the western blots of Figs. 2c and 3c, Supplementary Figs. 1b and 15b. Exact $p$ values are also included in this file. All biological material described in this study is available upon request. Source data are provided with this paper.

## Code availability

The code used to infer the GRNs in this study is available at https://github.com/arendma/GRN_code[85].

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

## Acknowledgements

We are grateful to professor Hideya Fukuzawa for sending us *lcr1*, *lcr1-C*, and their respective WT strain and to Professor Peter Jahns for the antibody against PSBS. We thank Michael P. Edwards and Qiao Wen for their feedback and contribution to GRN_code github repository. We would like to express our gratitude towards Arne Neumann for implementing the R shiny application. The authors would like to thank the following agencies for funding: The Human Frontiers Science Program through the funding of the project RGP0046/2018 (D.P., Z.N.); the French National Research Agency in the framework of the Young Investigators program ANR-18-CE20-0006 through the funding of the project MetaboLight (D.P.); the French National Research Agency in the framework of the Investissements d'Avenir program ANR-15-IDEX-02, through the funding of the "Origin of Life" project of the Univ. Grenoble-Alpes (D.P., Y.Y.); the French National Research Agency through the funding of the Grenoble Alliance for Integrated Structural & Cell Biology GRAL project ANR-17-EURE-0003 (D.P., M.A.R.-S.), the Prestige Marie-Curie co-financing grant PRESTIGE-2017-1-0028 (M.A.R.-S.); the International Max Planck Research School 'Primary Metabolism and Plant Growth' at the Max Planck Institute of Molecular Plant Physiology (M.A., Z.N.).

## Author contributions

M.A., Y.Y., M.A.R.-S. performed experiments. M.A., Y.Y., D.P. analyzed and visualized data. M.A. implemented consensus GRN inference. M.A., Y.Y., N.O., Z.N., D.P. designed study and planned research. M.A., Y.Y., Z.N., D.P. wrote the manuscript.

## Funding

## Competing interests

The authors declare no competing interests.
