## [Peer Review File · Nature Communications]

Widening the landscape of transcriptional regulation of green algal photoprotectionREVIEWER COMMENTS

Reviewer #3 (Remarks to the Author):

In this study, authors generated different GRNs using multiple approaches, evaluated them based on their predictions of known interactions and combined different GRNs (averaging ranks in individual networks) into a more robust network (consensus network).

Major comments:

Much of the evaluation of GRN performance draws from figure 2b, a heatmap of the interaction regulatory strengths (based on edge ranks) of known interactions inferred from each GRNs where colors were used to represent ranks (as a percent of total possible interactions / all*all) and grey indicate no edges. However, I find it hard to compare the networks due to several missing information; (i) the total number of edges of each GRNs were not shown and (ii) it is unclear if the lower limit of color scale (most red) is the worst possible ranking. Without this information it is hard to compare predictions between GRNs. Case in point, GENIE3 vs CLR: GENIE3 might look better than CLR because “reds” seemingly suggests a stronger regulatory strength than “greys” but this may not be true across networks.

In a situation where GENIE3 infers significantly more edges than CLR (say, 90% vs 50%), statistically speaking, “greys” in CLR are stronger than “reds” in GENIE3. I am putting much emphasis on the selection of best performing GRN method because GENIE3 is selected over CLR to construct the PHOT-specific GRN in this study and I am curious about the way “performance” is measured. To improve the heatmap in 2b, I suggest including in the number of recovered edges (in percentage of all possible edges) for each GRN.

The authors use SOR1-> PSBS1 interaction (shown experimentally to be absent) as a negative control to evaluate the performance of GRNs. However, I think that the statistical significance of this information is limited (or non-existent) based on only one observation. To have a better idea of the false positive rates of the different GRNs, it will be better to include more negative controls to make a statistical statement.

Regarding the genes of LHCSR3 (LHCSR3.1 and LHCSR3.2) and PSBS (PSBS1 and PSBS2). In results, the authors show the effect of LCR1 and QER7 in the expression levels of LHCSR3.1 and PSBS1. It will be interesting to also observe the effects in the genes LHCSR3.2 and PSBS2 or at least discuss why the authors are not showing the results of those genes.

Minor comments:

Line 48: ‘is encoded as’ -> ‘is encoded by’

Line 49-50: ‘...and accumulate..’ genes don’t accumulate. Please rewrite this sentence.

Line 80-85: It is great that the authors have pointed out the issue of collinearity due to limited number of samples. However, the authors must clarify that the consensus approach employed here will not in any way reduce collinearity of transcript levels despite increasing “robustness” of the GRN. Collinearity is a feature of a small transcriptomic dataset and should remain a concern for the authors; the RNA-seq compendium of only 158 samples was used to predict interactions between 407 TFs. I suggest including collinearity as a legitimate concern in the discussion.

In Figure 1a, please explain the order of the samples in the legend.

The title is very general since they study the process only in *C. reinhardtii*.

Line 97-98: Suggest using “experimentally validating” instead for clarity.

Line 106: Suggest changing “here generated” to “in-house generated”

Line 136: I think “Furthermore” should be used here instead of “Further”. Same goes for line 178 and line 82.

Line 141: I think “falsified” does not confer the meaning intended by the authors. Please change sentence to “moreover, interaction (shown to be absent in experiments) between SOR1 and PSBS1 was correctly discarded by all approaches”

Line 164: The subject of this sentence should be *lcr1*, the mutant and not LCR1, the protein.

Line 189-184: Authors must make it exceedingly clear that the PHOT-specific network is an amalgamation of the consensus network (inferred using whole RNA-seq compendium) and a GENIE3 GRN (inferred using a phot-specific dataset). Also, please use “borrowing” instead of “burrowing”. I think it is also important for authors to elaborate on how combining the two networks borrows “the statistical power of the RNA-seq compendium” as this is the justification of the chosen strategy. I think this approach might be used to prevent the pitfalls (mainly false positives due to transcript level collinearity) of a single GRN inferred from a small subset of samples, but I cannot be sure. I am sure readers will be interested to know.

Figure 2: The annotation of heatmap in panel A can be improved. From my understanding, there are 158 samples or 62 groups of replicates, each subjected to different permutations of conditions and diurnal sampling time points. However, it is difficult to map expression values to their respective experimental condition based on the limited number of features in the column header bars. This not only precludes readers from following the results inferred from the data laid out in the manuscript but also from identifying patterns and trends in gene expression. It is also not clear how the samples were arranged or if clustering is involved (as is commonly shown in expression heatmaps). Replicate relationships between samples are also absent in the heatmap so it is also difficult to access the replicability of RNA-seq dataset.

Figure 5: Please change “curate” to “curated” in legend. Also, it is unclear how regulatory strength of TF nodes (node size) are calculated from strength of interaction (edge width). Is it the sum / arithmetic mean of interaction strengths (edge weights) or is it the number of interactions (degree, in network terms)? Please clarify as all these can dramatically change how the network visualisation in figure 5 should be interpreted.

Reviewer #4 (Remarks to the Author):

This manuscript reports on the transcriptional regulation of photoprotective mechanisms and CO₂ concentrating mechanism (CCM) by generating a gene regulatory network (GRN) using 158 RNA seq data. Using consensus GRN calculated from several GRNs, the authors revealed that well-known CCM regulator LCR1 plays a role in the regulation of qE-related genes. They identified a novel qE regulator, QER7, using PHOT-specific GRN. The authors provided evidence that QER7 is a suppressor of non-photochemical quenching (NPQ) pathways by measuring the expression levels of qE-related genes and NPQ levels in the *qer7* mutant and its complementation strain. Furthermore, they proposed the possibility that blue light signaling is involved in the regulation of CCM gene expression by measuring expression levels of CCM genes in the *qer7* mutant and expression levels of QER7 in the phot mutant. This work was initiated with bioinformatic analyses to identify key regulators of NPQ and CCM. There are a variety of interesting observations and several novel findings, but some of their findings are not convincing. I have listed my concerns below.

Major comments

1. The authors' definition of representative CCM and qE-related genes is subtle. For example, the authors included PSBS1 and PSBS2 genes in qE-related genes and monitored their expression for testing induction of NPQ, although the role of PSBS1 and PSBS2 in photoprotective mechanisms are largely unknown in *Chlamydomonas* as mentioned in the introduction. Also, their NPQ measurement and qPCR results of qE-related genes in *lcr1* suggest that the role of PSBS1 in the induction of NPQ is marginal in their experimental conditions (Fig. 2b); both protein level and transcript level of PSBS increased while those of LHCSR3.1 decreased, resulting in the reduction of NPQ and qE in *lcr1* mutant. In CCM genes, it is difficult to understand why the authors included CAH1 and CCP1 in representative CCM genes and why they examined different CCM genes in Fig. 1 and Fig. 4 for examining the regulation of induction of CCM genes. The role of CCP1 in CCM is subtle (Pollock, Steve V., et al. "The *Chlamydomonas reinhardtii* proteins Ccp1 and Ccp2 are required for long-term growth, but are not necessary for efficient photosynthesis, in a low-CO₂ environment." *Plant molecular biology* 56.1 (2004): 125-132.) and several studies indicated that CAH1 is not directly involved in CCM in *Chlamydomonas* (Moroney, James V., et al. "The carbonic anhydrase isoforms of *Chlamydomonas reinhardtii*: intracellular location, expression, and physiological roles." *Photosynthesis Research* 109.1 (2011): 133-149.). It is necessary for explaining the reasonable reasons why these genes are listed in representative qE genes and CCM genes.

2. The authors constructed seven GRNs using different methods and verify the quality of GRNs using a single negative control (SOR1->PSBS1). They then decided to exclude two of them (ARACNE and Silencing) to make the consensus GRN in Fig.1, since GRNs from ARACNE and Silencing cannot reproduce TF-target interactions. The criteria for excluding GRNs are not clear. The authors need to explain why they use SOR1->PSBS1 as a single negative control for verifying the quality of GRNs. Also, it is required to include more negative controls for evidence to exclude ARACNE and Silencing methods from constructing the consensus GRN.

3. The data presented in this manuscript do not strongly support that QER7 represses the CCM. The only measurement of expression levels of CCM genes using qPCR in Fig.4 is not enough to claim that QER7 is a novel repressor of CCM. The measurement of inorganic carbon (Ci) affinity in *qer7* mutant and its complementation strain will be required to support your argument. Similarly, changes in QER7 transcript levels in phot mutant and its complementation strain (Fig. 4b) are not sufficient to state that blue-light perception via PHOT is involved in the regulation of CCM. If blue-light perception via PHOT is involved in the regulation of CCM, examination of expression levels of CCM genes and Ci affinity in phot will support your hypothesis. There is also no evidence showing that the perception of blue light is involved in the regulation of CCM. To the best of my knowledge, the role of blue light in CCM induction is marginal in *Chlamydomonas* (Borodin, V. B. "Effect of red and blue light on acclimation of *Chlamydomonas reinhardtii* to CO₂-limiting conditions." *Russian Journal of Plant Physiology* 55.4 (2008): 441-448.).

4. The discussion section in this manuscript is impoverished in supporting their findings and arguments. There are many interesting questions to be discussed in this manuscript. For example, if *Chlamydomonas* cells perceive blue light signals via PHOT and QER7 to regulate CCM, why blue light-mediated signaling networks is required for cells to induce CCM under high light conditions? Also, it is necessary to discuss the role of CCM induction in mitigating excess light energy under high light stress conditions. Furthermore, what is the physiological meaning of co-regulation of NPQ and CCM by QER7? It has been suggested that qE is an early time response under high light conditions to dissipate the excess light energy (Erickson, Erika, Setsuko Wakao, and Krishna K. Niyogi. "Light stress and photoprotection in *Chlamydomonas reinhardtii*." *The Plant Journal* 82.3 (2015): 449-465.). However, it is suggested that CCM-mediated enhancement of photochemical quenching can contribute to the dissipation of excess energy from light (Choi, Bae Young, et al. "The *Chlamydomonas* bZIP transcription factor BLZ8 confers oxidative stress tolerance by inducing the carbon-concentrating mechanism." *The Plant Cell* 34.2 (2022): 910-926.) and photochemical quenching requires time to be fully activated (Saroussi, Shai, et al.

"Alternative outlets for sustaining photosynthetic electron transport during dark-to-light transitions." *Proceedings of the National Academy of Sciences* 116.23 (2019): 11518-11527.). It would be very interesting to discuss why two temporally different stress responses are co-regulated by the QER7 regulator. I strongly recommend that the authors should rewrite the discussion section intensively.

Minor comments

1. Line 209-210, there was no immunoblot data of LHCSR1 in the extended data Fig.7.
2. Why is PHOT not involved in the top 10 regulators of qE-related genes in the consensus GRN (Fig. 2a), since it is one of the key regulators in NPQ induction?
3. How can the suppressor of qE genes, QER7, be involved in the top 10 regulators with the strongest cumulative regulatory effect on qE genes?
4. Line 242-244, what are the top 10 regulators of the CCM genes probed by qPCR and where is qPCR data for these 10 regulators, and how these 10 genes are selected? In Fig. 4, there are qPCR experiments for CCM genes but not for 10 regulators of the CCM genes. I cannot clearly understand the meaning of this sentence.
5. The figure legend in Fig. 4 needs more information about cell culture conditions. Since exogenous acetate and CO₂ concentration in the culture medium can highly alter the expression levels of CCM genes, it is better to explain the detailed culture conditions such as the existence of acetate in medium and air bubbling conditions or agitation information.
6. The last paragraph (Line 253-262) should constitute an independent result subsection.
7. Many minor typos (Fig. 4 legends, line 407 'was', and Line 166-167 and line 168-170 repeat of the same sentence, and so on). Please carefully rewrite the manuscript.

AUTHOR REBUTTALS TO REVIEWERS' COMMENTS:

We thank the reviewers for their time in reviewing this manuscript and their positive comments on the novelty and importance of this work. We also thank the reviewers for the constructive comments and suggestions that have helped us improve this manuscript.

In summary, we have made the following changes and additions:

- **Analysis of inference performance:** We repeated the analysis of inference performance with four additional negative regulatory interactions, that have been experimentally shown to be absent (updated Fig. 1b, added Supplementary Note 1).
- **Selection of inference approaches:** We substantially revised the respective results text section giving sensitivity of 0 as discrete exclusion criteria.
- **LHCSR3 and PSBS isoforms:** We analyzed the co-expression in our data set.
- **Information on RNA-seq conditions:** We updated Fig. 1a and Supplementary Table 1 and created a more detailed representation of Fig. 1a as Extended Data Fig. 1 to give additional information.
- **CCM genes and regulator predictions:** We updated the list of CCM genes considering reviewer remarks and rerun the regulator predictions. We substantially rewrote the result section referring to regulator prediction of CCM genes quantified by qPCR (updated Fig. 1a, Extended Data Fig. 11, Fig. 5).
- **LCR1- and QER7-mediated pathways regulating qE and CCM:** We analyzed the *cia5* mutant in terms of *QER7* gene expression (Fig. 4c). Together with the expression levels of *QER7* and *LCR1* in the *phot* mutant (Fig. 4b) we can conclude that *QER7* is repressed by PHOT and is CIA5-independent, while expression of *LCR1* is regulated by CIA5 and is PHOT-independent.
- **PHOT and CCM gene expression:** We extended our analyses to address whether PHOT contributes to the regulation of CCM gene expression and found that PHOT partially does contribute to this regulation (Supplementary Fig 13).
- **QER7, PHOT and affinity for inorganic carbon:** We performed experiments to address the role of *qer7* and *phot* mutants in CCM by measuring photosynthetic O₂ evolution as a function of the concentration of external inorganic carbon. Supplementary Figs 12 and 14, Supplementary Table 7. Results and Discussion sections have been modified accordingly.
- **Data Accessibility:** We developed an open-source webtool for easy query of the consensus and PHOT-Network predictions, hosted on a public server. Discussion and Data availability section have been augmented accordingly.

A point-by-point response to the reviewers' comments is presented below, in blue.

REVIEWER COMMENTS

Reviewer #3 (Remarks to the Author):

In this study, authors generated different GRNs using multiple approaches, evaluated them based on their predictions of known interactions and combined different GRNs (averaging ranks in individual networks) into a more robust network (consensus network).

Major comments:

Much of the evaluation of GRN performance draws from figure 2b, a heatmap of the interaction regulatory strengths (based on edge ranks) of known interactions inferred from each GRNs where colors were used to represent ranks (as a percent of total possible interactions / all*all) and grey indicate no edges. However, I find it hard to compare the networks due to several missing information; (i) the total number of edges of each GRNs were not shown and (ii) it is unclear if the lower limit of color scale (most red) is the worst possible ranking. Without this information it is hard to compare predictions between GRNs. Case in point, GENIE3 vs CLR: GENIE3 might look better than CLR because “reds” seemingly suggests a stronger regulatory strength than “greys” but this may not be true across networks. In a situation where GENIE3 infers significantly more edges than CLR (say, 90% vs 50%), statistically speaking, “greys” in CLR are stronger than “reds” in GENIE3. I am putting much emphasis on the selection of best performing GRN method because GENIE3 is selected over CLR to construct the PHOT-specific GRN in this study and I am curious about the way “performance” is measured. To improve the heatmap in 2b, I suggest including in the number of recovered edges (in percentage of all possible edges) for each GRN.

Response: We thank reviewer #3 for pointing out these improvements to the presentation of the performance analysis and we addressed the raised points in the redesigned Fig. 1b. We included the number of inferred edges per approach and clarified in the legend and figure caption that the color scale is independent of this number.

The authors use SOR1-> PSBS1 interaction (shown experimentally to be absent) as a negative control to evaluate the performance of GRNs. However, I think that the statistical significance of this information is limited (or non-existent) based on only one observation. To have a better idea of the false positive rates of the different GRNs, it will be better to include more negative controls to make a statistical statement.

Response: We addressed this request by reviewer #3 by adding four experimentally invalidated interactions to our ground truth data. These negative interactions are based on experiments with transcription factor knock-out or overexpressing mutants conducted in our laboratory during the course of other research projects. We would like to note that many regulatory interactions are only observable under specific conditions; thus, it is easier to experimentally confirm than to invalidate a regulatory interaction.

Regarding the genes of *LHCSR3* (*LHCSR3.1* and *LHCSR3.2*) and *PSBS* (*PSBS1* and *PSBS2*). In results, the authors show the effect of LCR1 and QER7 in the expression levels of *LHCSR3.1* and *PSBS1*. It will be interesting to also observe the effects in the genes *LHCSR3.2* and *PSBS2* or at least discuss why the authors are not showing the results of those genes.

Response: Reviewer#3 raises an important question. As we mention in our introduction (lines 47-49) the *LHCSR3.1* and *LHCSR3.2* genes in *Chlamydomonas* encode identical LHCSR3 proteins (Peers et al., 2009), while *PSBS1* and *PSBS2* encode PSBS proteins that differ only in one amino acid of the chloroplast transit peptide (Correa-Galvis et al., 2016). Both pairs of genes (*LHCSR3.1/2* and *PSBS1/2*) have been reported to yield highly similar transcript profiles (Redekop et al., 2022). Nevertheless, we checked the Pearson correlation between the pairs of paralogous genes over the 158 samples in our data set and confirmed that they have highly similar profiles (*LHCSR3.1* and *LHCSR3.2* : 0.98, *PSBS1* and *PSBS2* : 0.96). Therefore, we are convinced that experimental validation of the transcript level changes of *LHCSR3.1* is sufficient to

capture transcript level changes of both *LHCSR3.1* and *LHCSR3.2*; same is true for the use of *PSBS1* to probe for *PSBS1* and *PSBS2* transcript level changes.

For clarity, we added the following sentence in the section “Consensus GRN pinpoints LCR1 as a regulator of qE-related genes” of the Results (Lines 176-178):

“Since both the paralogs of *PSBS* as well as of *LHCSR3* show correlation greater than 0.96 over all RNAseq samples used in this study and additionally show very similar expression profiles quantified by RT-qPCR (Redekop et al., 2022), we only probed the transcripts of *LHCSR3.1* and *PSBS1* via qPCR in the validation assays”

Minor comments:

Line 48: ‘is encoded as’ -> ‘is encoded by’

Line 49-50: ‘...and accumulate..’ genes don’t accumulate. Please rewrite this sentence.

Response: We addressed these two points in the revised version of the manuscript.

Line 80-85: It is great that the authors have pointed out the issue of collinearity due to limited number of samples. However, the authors must clarify that the consensus approach employed here will not in any way reduce collinearity of transcript levels despite increasing “robustness” of the GRN. Collinearity is a feature of a small transcriptomic dataset and should remain a concern for the authors; the RNA-seq compendium of only 158 samples was used to predict interactions between 407 TFs. I suggest including collinearity as a legitimate concern in the discussion.

Response: We thank reviewer #3 for bringing up this important concern. We agree that the creation of a consensus network itself does not mitigate the problem of collinearity. However, by selecting inference approaches that employ different concepts of shrinkage (e. g. elastic net regression) or bagging (e. g. GENIE3) we indeed reduce artifacts originating from collinearity. This, however, does not fully exclude inference errors resulting from collinearity and we followed the reviewer’s advice to explicitly add this topic in the discussion.

In Figure 1a, please explain the order of the samples in the legend.

Response: This point was addressed in the revised Fig. 1.

The title is very general since they study the process only in *C. reinhardtii*.

Response: We made the title less broad by changing “algal to “green algal”. Now the title reads “Widening the landscape of transcriptional regulation of green algal photoprotection”.

Line 97-98: Suggest using “experimentally validating” instead for clarity.

Response: Validating was changed to “experimentally validating” as proposed by the reviewer (now appearing in lines 100, 136, 327, 354).

Line 106: Suggest changing “here generated” to “in-house generated”

Response: Modified as suggested, now appearing in line 109.

Line 136: I think “Furthermore” should be used here instead of “Further”. Same goes for line 178 and line 82.

Response: Modified as suggested.

Line 141: I think “falsified” does not confer the meaning intended by the authors. Please change sentence to “moreover, interaction (shown to be absent in experiments) between SOR1 and PSBS1 was correctly discarded by all approaches”

Response: Entire section was reworked to address this and other reviewer comments.

Line 164: The subject of this sentence should be lcr1, the mutant and not LCR1, the protein.

Response: Modified as suggested.

Line 189-184: Authors must make it exceedingly clear that the PHOT-specific network is an amalgamation of the consensus network (inferred using whole RNA-seq compendium) and a GENIE3 GRN (inferred using a phot-specific dataset). Also, please use “borrowing” instead of “burrowing”. I think it is also important for authors to elaborate on how combining the two networks borrows “the statistical power of the RNA-seq compendium” as this is the justification of the chosen strategy. I think this approach might be used to prevent the pitfalls (mainly false positives due to transcript level collinearity) of a single GRN inferred from a small subset of samples, but I cannot be sure. I am sure readers will be interested to know.

Response: We substantially rewrote and extended this section to address the valuable comments made by the reviewer.

Figure 2: The annotation of heatmap in panel A can be improved. From my understanding, there are 158 samples or 62 groups of replicates, each subjected to different permutations of conditions and diurnal sampling time points. However, it is difficult to map expression values to their respective experimental condition based on the limited number of features in the column header bars. This not only precludes readers from following the results inferred from the data laid out in the manuscript but also from identifying patterns and trends in gene expression. It is also not clear how the samples were arranged or if clustering is involved (as is commonly shown in expression heatmaps). Replicate relationships between samples are also absent in the heatmap so it is also difficult to access the replicability of RNA-seq dataset.

Response: We addressed these points during the revision of Fig. 1. To allow for a detailed inspection of expression values we added Extended Data Fig. 1 in the revised version of the Manuscript, that includes an enlarged version of the Fig. 1a along with sample names. We updated Supplementary Table 1 to allow mapping of the additional information given in the table to the heatmap in Extended Data Fig. 1.

Figure 5: Please change “curate” to “curated” in legend. Also, it is unclear how regulatory strength of TF nodes (node size) are calculated from strength of interaction (edge width). Is it the sum / arithmetic mean of interaction strengths (edge weights) or is it the number of interactions (degree, in network terms)? Please clarify as all these can dramatically change how the network visualisation in figure 5 should be interpreted.

Response: We revised the caption of Fig. 5 to make sure the mentioned information is clearly accessible.

Reviewer #4 (Remarks to the Author):

This manuscript reports on the transcriptional regulation of photoprotective mechanisms and CO₂ concentrating mechanism (CCM) by generating a gene regulatory network (GRN) using 158 RNA seq data. Using consensus GRN calculated from several GRNs, the authors revealed that well-known CCM regulator LCR1 plays a role in the regulation of qE-related genes. They identified a novel qE regulator, QER7, using PHOT-specific GRN. The authors provided evidence that QER7 is a suppressor of non-photochemical quenching (NPQ) pathways by measuring the expression levels of qE-related genes and NPQ levels in the *qer7* mutant and its complementation strain. Furthermore, they proposed the possibility that blue light signaling is involved in the regulation of CCM gene expression by measuring expression levels of CCM genes in the *qer7* mutant and expression levels of QER7 in the *phot* mutant. This work was initiated with bioinformatic analyses to identify key regulators of NPQ and CCM. There are a variety of interesting observations and several novel findings, but some of their findings are not convincing. I have listed my concerns below.

Major comments

1. The authors' definition of representative CCM and qE-related genes is subtle. For example, the authors included PSBS1 and PSBS2 genes in qE-related genes and monitored their expression for testing induction of NPQ, although the role of PSBS1 and PSBS2 in photoprotective mechanisms are largely unknown in *Chlamydomonas* as mentioned in the introduction. Also, their NPQ measurement and qPCR results of qE-related genes in *lcr1* suggest that the role of PSBS1 in the induction of NPQ is marginal in their experimental conditions (Fig. 2b); both protein level and transcript level of PSBS increased while those of LHCSR3.1 decreased, resulting in the reduction of NPQ and qE in *lcr1* mutant.

Response: We thank the reviewer #4 for these valuable comments. PSBS is the qE effector protein in higher plants, where it is constitutively expressed. While the mechanism of qE is conserved in green algae, the key qE protein effector is the LHCSR3. However, the presence of the two PSBS genes in the genome of *Chlamydomonas* and the fact that for several years researchers had not found conditions triggering PSBS protein accumulation (Bonente et al., 2008) created a long-standing problem about the role of PSBS in *Chlamydomonas*. We now know that PSBS protein does accumulate under very high light (transiently) or UV-B from the works of the labs of Peter Jahns (Correa-Galvis et al., 2016), Stefano Caffari (Tibiletti et al., 2016) and Michel Goldschmidt-Clermont (Allorent et al., 2016). Despite the fact that the precise contribution of PSBS in *Chlamydomonas*' photoprotective responses is still partly unresolved, we do know that PSBS does have a photoprotective role. Recent work from Peter Jahns' lab showed that PSBS proteins contribute to photoprotection during high light (HL) acclimation of *Chlamydomonas* through NPQ-independent and NPQ-dependent mechanisms (Redekop et al., 2020). We are therefore of the opinion that PSBS1/2 genes need to be included in our list of representative qE genes. We added the following sentence in the introduction (Lines 51-53) to clarify the photoprotective role of *Chlamydomonas* PSBS: "while their precise contribution in *Chlamydomonas*' photoprotective responses is still a matter of ongoing research, current understanding is that PSBS proteins contribute to photoprotection during HL acclimation of *Chlamydomonas* through NPQ-independent and NPQ-dependent mechanisms (Redekop et al., 2020)".

As reviewer #4 very correctly pointed out, overaccumulation of PSBS mRNA and protein levels (Fig. 2b-d) did not lead to sustainable NPQ in the absence of LHCSR3. This is in line with published data demonstrating that while downregulation of PSBS leads to decreased NPQ, overaccumulation of PSBS does not enhance NPQ (Redekop et al. 2020). Lastly, our finding that LCR1 acts as a repressor of PSBS protein accumulation is expected to advance our understanding of the labile nature of PSBS protein accumulation under HL in *Chlamydomonas* (Correa-Galvis et al., 2016; Redekop et al., 2020).

In CCM genes, it is difficult to understand why the authors included CAH1 and CCP1 in representative CCM genes and why they examined different CCM genes in Fig. 1 and Fig. 4 for examining the regulation of induction of CCM genes. The role of CCP1 in CCM is subtle (Pollock, Steve V., et al. "The Chlamydomonas reinhardtii proteins Ccp1 and Ccp2 are required for long-term growth, but are not necessary for efficient photosynthesis, in a low-CO₂ environment." *Plant molecular biology* 56.1 (2004): 125-132.) and several studies indicated that CAH1 is not directly involved in CCM in Chlamydomonas (Moroney, James V., et al. "The carbonic anhydrase isoforms of Chlamydomonas reinhardtii: intracellular location, expression, and physiological roles." *Photosynthesis Research* 109.1 (2011): 133-149.). It is necessary for explaining the reasonable reasons why these genes are listed in representative qE genes and CCM genes.

Response: We thank the reviewer for raising this point. Our reasoning for including CCP1 and CAH1 in the list of representative CCM genes was that independently of their physiological implication in CCM, expression of those genes is under the control of the CIA5, the master regulation of CCM and low CO₂-responsive genes (Pollock et al., 2004; Xiang et al., 2001; Fukuzawa et al., 2001). To improve clarity, we removed the two genes from the list of CCM genes and from Fig. 1a and Fig. 4a. Additionally, in the revised version of the manuscript, Fig. 1a, 4a and Extended Data Fig. 8 now illustrate the same subset of CCM genes.

2. The authors constructed seven GRNs using different methods and verify the quality of GRNs using a single negative control (SOR1->PSBS1). They then decided to exclude two of them (ARACNE and Silencing) to make the consensus GRN in Fig.1, since GRNs from ARACNE and Silencing cannot reproduce TF-target interactions. The criteria for excluding GRNs are not clear. The authors need to explain why they use SOR1->PSBS1 as a single negative control for verifying the quality of GRNs. Also, it is required to include more negative controls for evidence to exclude ARACNE and Silencing methods from constructing the consensus GRN.

Response: We thank reviewer #4 for pointing out the need to make the decision on which networks to exclude more comprehensible. We substantially rewrote the respective part of the result section and made clear that ARACNE and silencing were excluded based on their sensitivity of 0%.

Following the required increase in the number of negative regulatory interactions in our ground truth data, we also added four additional experimentally confirmed interactions based on experiments with transcription factor knock-out or overexpressing strains conducted in our laboratory during the course of other research projects. Since ARACNE and Silencing were excluded based on their inability to infer experimentally confirmed interactions, the addition does not affect the selection of the approaches used to create the consensus network.

3. The data presented in this manuscript do not strongly support that QER7 represses the CCM. The only measurement of expression levels of CCM genes using qPCR in Fig.4 is not enough to claim that QER7 is a novel repressor of CCM. The measurement of inorganic carbon (Ci) affinity in qer7 mutant and its complementation strain will be required to support your argument. Similarly, changes in QER7 transcript levels in phot mutant and its complementation strain (Fig. 4b) are not sufficient to state that blue-light perception via PHOT is involved in the regulation of CCM. If blue-light perception via PHOT is involved in the regulation of CCM, examination of expression levels of CCM genes and Ci affinity in phot will support your hypothesis. There is also no evidence showing that the perception of blue light is involved in the regulation of CCM. To the best of my knowledge, the role of blue light in CCM induction is marginal in Chlamydomonas (Borodin, V. B. "Effect of red and blue light on acclimation of Chlamydomonas reinhardtii to CO₂-limiting conditions." *Russian Journal of Plant Physiology* 55.4 (2008): 441-448.).

Response: The reviewer raises two very interesting questions: (1) does the overexpression of CCM-related genes in *qer7* translate to elevated CCM as compared to the WT? (2) is PHOT involved in the regulation of

CCM? Motivated by the reviewer's comments and suggestions, we performed additional experiments to address both questions.

We compared WT and *qer7* cells for their affinity for Ci, under conditions where CCM is not fully induced (LL) and after inducing CCM by acclimation to HL, which leads to mRNA accumulation of CCM-related genes (see for example **Fig. 4a** but also our previous study (Ruiz-Sola et al., 2021)) and to accumulation of CCM proteins (Scholz et al., 2019). Under these conditions no difference could be observed between WT and *qer7* (**Supplementary Fig. 12, Supplementary Table 7**). Our data suggest that PHOT, by repressing *QER7* (**Fig. 4b**), is also involved in the regulation of CCM related gene expression. We investigated this further by quantifying CCM gene expression in WT, *phot* mutant and the complemented line *phot-C*, in the samples collected from the experiment presented in Fig. 4b, i.e. synchronized photoautotrophic cultures shifted to HL right after the end of the dark phase. We first analyzed expression of qE genes that was found as expected (Petroutsos et al., 2016; Redekop et al., 2022; Aihara et al., 2019) to be under control of PHOT (**Supplementary Fig. 13a**). We then analyzed expression of the five CCM genes found to be repressed by *QER7* (**Fig. 4a**); out of those, *CAH4* and *HLA3* were down-regulated in the *phot* mutant exposed to HL after the end of the dark phase in synchronized cultures (**Supplementary Fig. 13b**) and this phenotype was fully rescued in the *phot-C* complemented line, suggesting a potential involvement of PHOT in regulating expression of CCM-related genes. Nevertheless, the affinity of *phot* for Ci was not different than this of WT (**Supplementary Fig. 14, Supplementary Table 7**), in line with previous work reporting CCM to be induced to very similar extent under blue or red illumination (Borodin, 2008).

The CCM phenotype of *qer7* and *phot* mutants was also tested in cells acclimated to 5% CO₂ and then shifted to air. In agreement with our LL to HL experiments, the 5% CO₂ to air shift experiments did not reveal differences in the affinity to Ci. These results are not included in the revised manuscript.

Thus, under our experimental conditions, the role of PHOT and *QER7* in regulating CCM is restricted to the transcriptional level. Since the PHOT-*QER7* pathway acts independently of the LCR1 pathway, both regulating the same subset of CCM genes we tested, it may be not so surprising that neither PHOT nor *QER7* impact the affinity for Ci; in both *qer7* and *phot* mutants LCR1 levels are unaffected and therefore the control of LCR1 on CCM may mask any potential effect that PHOT or *QER7* might have. Since many known CCM regulatory mechanisms act posttranscriptionally (Santhanagopalan et al., 2021), it is conceivable, that the transcriptional regulation of PHOT and *QER7* on their own are not sufficient and rely on integration with other simultaneous signals to clear all roadblocks for full CCM induction under HL.

These new results are discussed both in the results and discussion sections.

4. The discussion section in this manuscript is impoverished in supporting their findings and arguments. There are many interesting questions to be discussed in this manuscript. For example, if *Chlamydomonas* cells perceive blue light signals via PHOT and *QER7* to regulate CCM, why blue light-mediated signaling networks is required for cells to induce CCM under high light conditions? Also, it is necessary to discuss the role of CCM induction in mitigating excess light energy under high light stress conditions. Furthermore, what is the physiological meaning of co-regulation of NPQ and CCM by *QER7*? It has been suggested that qE is an early time response under high light conditions to dissipate the excess light energy (Erickson, Erika, Setsuko Wakao, and Krishna K. Niyogi. "Light stress and photoprotection in *Chlamydomonas reinhardtii*." *The Plant Journal* 82.3 (2015): 449-465.). However, it is suggested that CCM-mediated enhancement of photochemical quenching can contribute to the dissipation of excess energy from light (Choi, Bae Young, et al. "The *Chlamydomonas* bZIP transcription factor BLZ8 confers oxidative stress tolerance by inducing the carbon-concentrating mechanism." *The Plant Cell* 34.2 (2022): 910-926.) and photochemical quenching requires time to be fully activated (Saroussi, Shai, et al. "Alternative outlets for sustaining photosynthetic electron transport during dark-to-light transitions." *Proceedings of the National Academy of Sciences* 116.23 (2019): 11518-11527.). It would be very interesting to discuss why two temporally

different stress responses are co-regulated by the QER7 regulator. I strongly recommend that the authors should rewrite the discussion section intensively.

Response: We substantially modified the discussion to address the reviewer's valuable comments. The discussion now addresses the physiological meaning of the co-regulation of photoprotection and CCM and the interconnection of the two processes at the molecular level while putting into perspective our findings of LCR1/QER7 as co-regulators between CCM and photoprotection gene expression. The role of PHOT in the gene expression regulation of CCM is also discussed.

Minor comments

1. Line 209-210, there was no immunoblot data of LHCSR1 in the extended data Fig.7.

Response: Thank you for pointing this out. We removed the reference to protein levels from the text.

2. Why is PHOT not involved in the top 10 regulators of qE-related genes in the consensus GRN (Fig. 2a), since it is one of the key regulators in NPQ induction?

Response: The PHOT gene encodes a membrane light receptor that is not directly involved in transcriptional regulation and therefore is not a member of the set of TFs in the inferred GRNs. For further details on the criteria by which TFs have been selected we kindly refer to the Method section of the Manuscript and the cited gene homology studies (Pérez-Rodríguez et al., 2009; Jin et al., 2017).

3. How can the suppressor of qE genes, QER7, be involved in the top 10 regulators with the strongest cumulative regulatory effect on qE genes?

Response: The methods used to infer the GRNs are capable of inferring both activating and inhibiting regulatory interactions. The ranking is based solely on the confidence of the regulatory effect. The introduction and result sections have been updated to emphasize this information.

4. Line 242-244, what are the top 10 regulators of the CCM genes probed by qPCR and where is qPCR data for these 10 regulators, and how these 10 genes are selected? In Fig. 4, there are qPCR experiments for CCM genes but not for 10 regulators of the CCM genes. I cannot clearly understand the meaning of this sentence.

Response: We thank reviewer #4 for pointing out that this section was somehow unclear. We predicted regulators for the CCM genes included in Fig. 4a. Those predictions were made in the same way the regulators for qE genes were predicted, as discussed in the Methods. We rewrote the mentioned section to improve comprehensibility.

5. The figure legend in Fig. 4 needs more information about cell culture conditions. Since exogenous acetate and CO₂ concentration in the culture medium can highly alter the expression levels of CCM genes, it is better to explain the detailed culture conditions such as the existence of acetate in medium and air bubbling conditions or agitation information.

Response: The results from the experiments presented in Fig. 4 were obtained from photoautotrophic conditions (Sueoka's high salt medium; HSM), at 23 °C, in Erlenmeyer flasks shaken at 125 rpm. We added this information to the figure legend as suggested by the reviewer.

6. The last paragraph (Line 253-262) should constitute an independent result subsection.

Response: We followed this valuable suggestion of reviewer4 in the updated MS.

7. Many minor typos (Fig. 4 legends, line 407 'was', and Line 166-167 and line 168-170 repeat of the same sentence, and so on). Please carefully rewrite the manuscript.

Response: We addressed this in the rewriting process.

References

- Aihara, Y., Fujimura-Kamada, K., Yamasaki, T., and Minagawa, J.** (2019). Algal photoprotection is regulated by the E3 ligase CUL4-DDB1DET1. *Nature Plants* **5**: 34–40.
- Allorent, G., Lefebvre-Legendre, L., Chappuis, R., Kuntz, M., Truong, T.B., Niyogi, K.K., Ulm, R., and Goldschmidt-Clermont, M.** (2016). UV-B photoreceptor-mediated protection of the photosynthetic machinery in *Chlamydomonas reinhardtii*. *Proc. Natl. Acad. Sci. U.S.A.* **113**: 14864–14869.
- Bonente, G., Passarini, F., Cazzaniga, S., Mancone, C., Buia, M.C., Tripodi, M., Bassi, R., and Caffarri, S.** (2008). The occurrence of the psbS gene product in *Chlamydomonas reinhardtii* and in other photosynthetic organisms and its correlation with energy quenching. *Photochem. Photobiol.* **84**: 1359–1370.
- Borodin, V.B.** (2008). Effect of red and blue light on acclimation of *Chlamydomonas reinhardtii* to CO₂-limiting conditions. *Russian Journal of Plant Physiology* **55**: 441–448.
- Correa-Galvis, V., Redekop, P., Guan, K., Griess, A., Truong, T.B., Wakao, S., Niyogi, K.K., and Jahns, P.** (2016). Photosystem II Subunit PsbS Is Involved in the Induction of LHCSR Protein-dependent Energy Dissipation in *Chlamydomonas reinhardtii*. *J. Biol. Chem.* **291**: 17478–17487.
- Fukuzawa, H., Miura, K., Ishizaki, K., Kucho, K.-I., Saito, T., Kohinata, T., and Ohyama, K.** (2001). Ccm1, a regulatory gene controlling the induction of a carbon-concentrating mechanism in *Chlamydomonas reinhardtii* by sensing CO₂ availability. *Proc. Natl. Acad. Sci. U.S.A.* **98**: 5347–5352.
- Jin, J., Tian, F., Yang, D.-C., Meng, Y.-Q., Kong, L., Luo, J., and Gao, G.** (2017). PlantTFDB 4.0: Toward a central hub for transcription factors and regulatory interactions in plants. *Nucl. Acids Res.* **45**: D1040–D1045.
- Peers, G., Truong, T.B., Ostendorf, E., Busch, A., Elrad, D., Grossman, A.R., Hippler, M., and Niyogi, K.K.** (2009). An ancient light-harvesting protein is critical for the regulation of algal photosynthesis. *Nature* **462**: 518–521.
- Petroutsos, D., Tokutsu, R., Maruyama, S., Flori, S., Greiner, A., Magneschi, L., Cusant, L., Kottke, T., Mittag, M., Hegemann, P., Finazzi, G., and Minagawa, J.** (2016). A blue-light photoreceptor mediates the feedback regulation of photosynthesis. *Nature* **537**: 563–566.
- Pérez-Rodríguez, P., Riaño-Pachón, D.M., Corrêa, L.G.G., Rensing, S.A., Kersten, B., and Mueller-Roeber, B.** (2009). PlnTFDB: Updated content and new features of the plant transcription factor database. *Nucl. Acids Res.* **38**.
- Pollock, S.V., Prout, D.L., Godfrey, A.C., Lemaire, S.D., and Moroney, J.V.** (2004). The *Chlamydomonas reinhardtii* proteins Ccp1 and Ccp2 are required for long-term growth, but are not necessary for efficient photosynthesis, in a low-CO₂ environment. *Plant Mol. Biol.* **56**: 125–132.
- Redekop, P., Rothhausen, N., Rothhausen, N., Melzer, M., Mosebach, L., Dülger, E., Bovdilova, A., Caffarri, S., Hippler, M., and Jahns, P.** (2020). PsbS contributes to photoprotection in *Chlamydomonas reinhardtii* independently of energy dissipation. *Biochimica et Biophysica Acta (BBA) - Bioenergetics* **1861**: 148183.
- Redekop, P., Sanz-Luque, E., Yuan, Y., Villain, G., Petroutsos, D., and Grossman, A.R.** (2022). Transcriptional regulation of photoprotection in dark-to-light transition- more than just a matter of excess light energy. *Sci Adv.*
- Ruiz-Sola, M. et al.** (2021). Photoprotection is regulated by light-independent CO₂ availability. *bioRxiv.*

- Santhanagopalan, I., Wong, R., Mathur, T., and Griffiths, H.** (2021). Orchestral manoeuvres in the light: crosstalk needed for regulation of the *Chlamydomonas* carbon concentration mechanism. *J. Exp. Bot.* **72**: 4604–4624.
- Scholz, M., Gäbelein, P., Xue, H., Mosebach, L., Bergner, S.V., and Hippler, M.** (2019). Light-dependent N-terminal phosphorylation of LHCSR3 and LHCB4 are interlinked in *Chlamydomonas reinhardtii*. *Plant J.* **99**: 877–894.
- Tibiletti, T., Auroy, P., Peltier, G., and Caffarri, S.** (2016). *Chlamydomonas reinhardtii* PsbS Protein Is Functional and Accumulates Rapidly and Transiently under High Light. *Plant Physiol.* **171**: 2717–2730.
- Xiang, Y., Zhang, J., and Weeks, D.P.** (2001). The *Cia5* gene controls formation of the carbon concentrating mechanism in *Chlamydomonas reinhardtii*. *Proc. Natl. Acad. Sci. U.S.A.* **98**: 5341–5346.

REVIEWERS' COMMENTS

Reviewer #3 (Remarks to the Author):

The authors have addressed my comments.

Reviewer #4 (Remarks to the Author):

I appreciated that the authors have performed several new experiments to address my concerns. In the discussion, I suggested to discuss why two temporally different stress responses (NPQ, photochemical quenching mediated by CCM) are co-regulated by the QER7 regulator. Please add some discussion on this or explain why the authors exclude this point in the discussion section. In addition, please check typos (line 337 electron chain -> electron transport chain, line 342 BZL8-> BLZ8).

I think that the revised manuscript has been substantially improved. This revised manuscript looks good and I don't have any further comments.

Response to the reviewers' comments

We are grateful to both reviewers who invested time and effort in reviewing this manuscript. Their comments and suggestions helped us improved the overall presentation of our work.

Reviewer #3 (Remarks to the Author):

The authors have addressed my comments.

- We are happy to read that our revised manuscript addressed all the comments of the reviewer.

Reviewer #4 (Remarks to the Author):

I appreciated that the authors have performed several new experiments to address my concerns. In the discussion, I suggested to discuss why two temporally different stress responses (NPQ, photochemical quenching mediated by CCM) are co-regulated by the QER7 regulator. Please add some discussion on this or explain why the authors exclude this point in the discussion section.

- We are pleased to read that the revised manuscript addressed the reviewer's previous concerns and we thank the reviewer for raising this interesting point. We decided not to include this in our MS because from the regulation perspective exposure to excess light leads to depletion of CO₂ and activation of both qE and CCM genes¹ and proteins², suggesting that the two processes are not temporally separated and that they are strongly interconnected. Indeed, this is evidenced by the fact that they share the QER7 and LCR1 regulators (from the present study) but also CIA5^{1,3-5}.

In addition, please check typos (line 337 electron chain -> electron transport chain, line 342 BZL8-> BLZ8).

- We have corrected those typos, thank you for pointing those out.

I think that the revised manuscript has been substantially improved. This revised manuscript looks good and I don't have any further comments.

References

1. Ruiz-Sola, M. Á., Flori, S., Yuan, Y., Villain, G., Sanz-Luque, E., Redekop, P., Tokutsu, R., Kueken, A., Tschla, A., Kepesidis, G., Allorent, G., Arend, M., Iacono, F., Finazzi, G., Hippler, M., Nikoloski, Z., Minagawa, J., Grossman, A. & Petroustos, D. Light-independent regulation of algal photoprotection by CO₂ availability. *Nature Communications* (2023, accepted).
2. Scholz, M., Gäbelein, P., Xue, H., Mosebach, L., Bergner, S. V. & Hippler, M. Light-dependent N-terminal phosphorylation of LHCSR3 and LHCB4 are interlinked in *Chlamydomonas reinhardtii*. *Plant J* 99, 877–894 (2019).
3. Miura, K., Yamano, T., Yoshioka, S., Kohinata, T., Inoue, Y., Taniguchi, F., Asamizu, E., Nakamura, Y., Tabata, S., Yamato, K. T., Ohyama, K. & Fukuzawa, H. Expression profiling-based identification of CO₂-responsive genes regulated by CCM1 controlling a carbon-

concentrating mechanism in *Chlamydomonas reinhardtii*. *Plant Physiol.* 135, 1595–1607 (2004).

4. Fang, W., Si, Y., Douglass, S., Casero, D., Merchant, S. S., Pellegrini, M., Ladunga, I., Liu, P. & Spalding, M. H. Transcriptome-wide changes in *Chlamydomonas reinhardtii* gene expression regulated by carbon dioxide and the CO₂-concentrating mechanism regulator CIA5/CCM1. *Plant Cell* 24, 1876–1893 (2012).

5. Redekop, P., Sanz-Luque, E., Yuan, Y., Villain, G., Petroustos, D. & Grossman, A. R. Transcriptional regulation of photoprotection in dark-to-light transition—More than just a matter of excess light energy. *Sci Adv* 8, eabn1832 (2022).